# The Adaptive Doubly Robust Estimator
# and a Paradox Concerning Logging Policy

**Masahiro Kato**[*]
CyberAgent, Inc.

**Kenichiro McAlinn**
Temple University

**Shota Yasui**
CyberAgent, Inc.

## Abstract

The *doubly robust* (DR) estimator, which consists of two nuisance parameters, the conditional mean outcome and the logging policy (the probability of choosing an action), is crucial in causal inference. This paper proposes a DR estimator for *dependent samples* obtained from adaptive experiments. To obtain an asymptotically normal semiparametric estimator from dependent samples with non-Donsker nuisance estimators, we propose *adaptive-fitting* as a variant of sample-splitting. We also report an empirical *paradox* that our proposed DR estimator tends to show better performances compared to other estimators utilizing the true logging policy. While a similar phenomenon is known for estimators with i.i.d. samples, traditional explanations based on asymptotic efficiency cannot elucidate our case with dependent samples. We confirm this hypothesis through simulation studies.

## 1 Introduction

*Adaptive experiments*, including efficient treatment effect estimation (van der Laan, 2008; Hahn et al., 2011), treatment regimes (Zhang et al., 2012a; Zhao et al., 2012), and multi-armed bandit (MAB) problems (Gittins, 1989; Lattimore & Szepesvári, 2020), are widely accepted and used in real-world applications, for example, in social experiments (Hahn et al., 2011), online advertisement (Zhang et al., 2012b), clinical trials (Chow & Chang, 2011; Villar, 2018), website optimization (White, 2012), and recommendation systems (Li et al., 2010). In those various applications, there is a significant interest in off-line counterfactual inference using samples obtained from past trials generated by a logging policy (the probability of choosing an action: Li et al., 2010, 2011). Among several off-line evaluation criteria, this paper focuses on *off-policy value estimation* (OPVE: Li et al., 2015). The goal of OPVE is to estimate the expected value of the weighted outcome, which includes average treatment effect (ATE) estimation as a special case (Hirano et al., 2003; Bang & Robins, 2005).

In OPVE, when the true logging policy is known, there are mainly three types of estimators: inverse probability weighting (IPW: Horvitz & Thompson, 1952), direct method (DM), and augmented IPW (AIPW Bang & Robins, 2005) estimators. IPW-type estimators refer to a sample average of the outcomes weighted by the true logging policy. DM-type estimators refer to a sample average of an estimated conditional outcome. AIPW-type estimators consist of two components: the IPW part and the DM part. More details are given in Section 3. In addition, when the true logging policy is unknown, we call IPW-type and AIPW-type estimators that use the estimated logging policy EIPW-type and DR-type estimators, respectively. In particular, and the focus of this paper, DR-type estimators are crucial in OPVE, since in most applications of interest the true logging policy is either unknown, difficult, or costly to obtain, making IPW and AIPW-type estimators often infeasible.

While existing OPVE methods typically assume that the samples are *independent and identically distributed* (i.i.d.: Hirano et al., 2003; Dudík et al., 2011), adaptive experiments usually allow logging policies to be updated based on past observations, under which the generated samples are non-i.i.d.

---

[*]`masahiro_kato@cyberagent.co.jp`

35th Conference on Neural Information Processing Systems (NeurIPS 2021).

In this case, theoretical results shown under i.i.d. assumptions, such as consistency and asymptotic normality, are not guaranteed. In particular, the asymptotic normality is critical, as we can obtain $\sqrt{T}$-rate confidence intervals, where $T$ is the sample size. For this problem, van der Laan (2008) proposed an asymptotically normal AIPW estimator under dependent samples, with several follow-up studies, including Luedtke & van der Laan (2016, 2018), Hadad et al. (2021), Kato et al. (2020), and Zhan et al. (2021). However, a DR-type estimator for dependent samples has not been formally proposed, despite its importance.

Readers might think that, in many experimental designs, the experiment can be manipulated by the experimenter, thus the information of the logging policy is always accessible. However, the logging policy is often not recorded. This is because it is costly to do so in adaptive experiments, since we need to preserve a policy for each sample. Consider the case of Thompson sampling, where we select the arm with the highest expected reward sampled from the posterior distribution (Thompson, 1933). In this case, if we want to know the logging policy, we need to calculate the probability that each arm is chosen from the posterior distribution. The calculation of this probability is not necessary for the bandit algorithm itself, but is an additional task for policy evaluation. In addition, to compute the logging policy, we not only need to record the information of the posterior distribution, but also need to calculate the probability by Monte Carlo simulation from the posterior distribution, if no analytical solution is available. This is a difficult task in terms of both computational volume and computational resource. Moreover, in adaptive experiments in industry (e.g. AB tests), the number of logs is often in the order of several million. In such cases, storing the logging policy is also a difficult task. By using DR-type estimators, we can alleviate this cost. Further, as we report below as a paradox, it is even possible that DR-type estimators, that do not use the true logging policy, outperform estimators using the true logging policy,

This paper proposes an asymptotically normal DR estimator for dependent samples. We call this DR estimator the *adaptive DR* (ADR) estimator. There are mainly two difficulties concerning this estimator. First, we cannot use the central limit theorem (CLT) for i.i.d. or martingale difference sequences directly, as done in van der Laan (2008), Hadad et al. (2021), and Kato et al. (2020). Second, when nuisance parameters are estimated with complicated models, the Donsker condition does not hold in general, which is required to show the asymptotic normality of semiparametric estimators, such as our proposed ADR estimator. In this paper, to solve these problems, we propose adaptive-fitting, inspired by the double/debiased machine learning for i.i.d. samples (Chernozhukov et al., 2018). We also find that by using the ADR estimator, not only can OPVE be done without knowing the true logging policy, the ADR estimator paradoxically often outperforms the performance of AIPW estimators that use the true logging policy. We list our contributions as follows.

**(I) Asymptotic normality of the ADR estimator.** We show the asymptotic normality of our proposed ADR estimator. To the best of our knowledge, a DR-type estimator for dependent samples obtained in adaptive experiments has not been formally proposed.

**(II) Adaptive fitting.** While the asymptotic normalities of IPW and AIPW estimators are shown by the martingale CLT, we cannot directly apply this technique to our proposed ADR estimator. This is because the martingale condition does not hold when the true logging policy is unknown. To solve this problem, we utilize a novel sample-splitting method called adaptive-fitting to show the asymptotic normality of the ADR estimator. We generalize this technique as a variant of sample-splitting (Chernozhukov et al., 2018) for semiparametric inference with dependent samples without the Donsker condition. Unlike existing sample-splitting methods, which assume that samples are i.i.d., the proposed adaptive-fitting is applicable to non-i.i.d. samples.

**(III) Empirical paradox on using estimated logging policy.** Through experimental studies, we investigate the empirical performance of our proposed ADR estimator. We find that the ADR estimator often exhibits improved performances over other estimators using the true logging policy, such as the AIPW estimator. Estimators requiring the true logging policy tend to empirically be unstable due to the instability of the nuisance parameter; the instability being caused by the logging policy being near zero before convergence. Our finding implies that the ADR estimator mitigates this instability. We call this phenomenon a *paradox concerning logging policy* because using more information (the true logging policy) does not improve empirical performance. A similar paradox is known for IPW-type estimators for i.i.d. samples (Hahn, 1998; Henmi & Eguchi, 2004). However, we cannot explain our paradox from a conventional semiparametric efficiency perspective (Hahn, 1998; Hirano et al., 2003),

as the asymptotic variances are the same between the ADR estimator (with an estimated logging policy) and the AIPW estimator (with the true logging policy), unlike IPW-type estimators.

**Organization of this paper.** In Section 2, we formulate our problem. In Section 3, we introduce existing OPVE estimators for dependent samples. In Section 4, we propose the ADR estimator, which is our main contribution. We call the method used for the ADR estimator adaptive-fitting, and generalize it in Section 5. In Section 6, we report a paradox through simulation studies and explain the cause of this phenomenon. In Section 7, we conduct experiments using benchmark datasets.

## 2 Problem setting

Suppose that there is a time series $1, 2, \ldots, T$, and denote the set as $[T] = \{1, \ldots, T\}$. For $t \in [T]$, let $A_t$ be an *action* in $\mathcal{A} = \{1, 2, \ldots, K\}$, $X_t$ be a *covariate* observed by a decision maker when choosing an action, and $\mathcal{X}$ be its space. Following the Neyman–Rubin causal model (Rubin, 1974), let a reward at period $t$ be $Y_t = \sum_{a=1}^{K} \mathbb{1}[A_t = a]Y_t(a)$, where $Y_t(a) : \mathcal{A} \to \mathbb{R}$ is a potential (random) outcome. In summary, we have a dataset $\{(X_t, A_t, Y_t)\}_{t=1}^{T}$ with the following data-generating process (DGP):

**Step 1:** we observe a covariate $X_t \sim p(x)$;

**Step 2:** we choose $A_t = a$ with probability $p_t(a|x)$;

**Step 3:** we observe an outcome $Y_t = \sum_{a=1}^{K} \mathbb{1}[A_t = a]Y_t(a)$, where $Y_t(a) \sim p_a(y|x)$,

where $p(x)$ denotes the density of the covariate $X_t$, $p_t(a|x)$ denotes the probability of choosing an action $a$ conditional on a covariate $x$ at period $t$, and $p_a(y|x)$ denotes the density of a reward $Y_t(a)$ conditional on a covariate $x$. We assume that $p(x)$ and $p_a(y|x)$ are invariant across periods, and $X_t$ and $Y_t(a)$ are drawn from these densities independently of the other period random variables; that is, $\{(X_t, Y_t(1), \ldots, Y_t(K))\}_{t=1}^{T}$ is i.i.d., but $p_t(a|x)$ can take different values across periods based on past observations. In this case, the samples $\{(X_t, A_t, Y_t)\}_{t=1}^{T}$ are correlated over time, that is, the samples are not i.i.d. Let $\Omega_{t-1} = \{X_{t-1}, A_{t-1}, Y_{t-1}, \ldots, X_1, A_1, Y_1\}$ be the history and $\mathcal{M}_{t-1}$ be a set of possible histories until the $t$-th period. The probability $p_t(a|x)$ is determined by a *logging policy* $\pi_t : \mathcal{A} \times \mathcal{X} \times \mathcal{M}_{t-1} \to (0, 1)$, such that $\sum_{a=1}^{K} \pi_t(a|x, \Omega_{t-1}) = 1$, which is a function of a covariate $X_t$, an action $A_t$, and history $\Omega_{t-1}$. We also assume that $\pi_t$ is conditionally independent of $Y_t(a)$ to satisfy unconfoundedness (Remark 2).

**Remark 1** (Stable unit treatment value assumption (SUTVA))**.** *The DGP implies SUTVA; that is, $p_a(y|x)$ is invariant to random assignments in other periods (Angrist et al., 1996).*

**Remark 2** (Unconfoundedness)**.** *In this paper, unconfoundedness refers to independence between $(Y_t(1), \ldots, Y_t(K))$ and $A_t$, conditional on $X_t$ and $\Omega_{t-1}$, which is required for identification.*

### 2.1 OPVE

The goal of OPVE is to estimate the expected value of the sum of the outcomes $Y_t(a)$ weighted by an evaluation function $\pi^{\mathrm{e}} : \mathcal{A} \times \mathcal{X} \to \mathbb{R}$; that is,

$$R(\pi^{\mathrm{e}}) := \mathbb{E}_{(X_t, Y_t(1), \ldots, Y_t(K))}\left[\sum_{a=1}^{K} \pi^{\mathrm{e}}(a|X_t)Y_t(a)\right],$$

where the expectation $\mathbb{E}_{(X_t, Y_t(1), \ldots, Y_t(K))}$ is taken over $(X_t, Y_t(1), \ldots, Y_t(K))$. We denote it as $\mathbb{E}$, when there is no ambiguity. As with Dudík et al. (2011), the evaluation function is usually referred to as an evaluation policy, where $\pi^{\mathrm{e}}(a|x) \in [0, 1]$ and the sum is 1. However, to not restrict it so we can include other forms, such as the ATE, we refer to it differently. The ATE is also a special case of OPVE for $\mathcal{A} = \{1, 2\}$, where $\pi^{\mathrm{e}}(1|x) = 1$ and $\pi^{\mathrm{e}}(2|x) = -1$. To identify $R(\pi^{\mathrm{e}})$, we assume the overlap in the distributions of policies, convergence of $\pi_{t-1}$, and the boundedness of reward.

**Assumption 1.** *For all $t \in [T]$, $a \in \mathcal{A}$, $x \in \mathcal{X}$, and $\Omega_{t-1} \in \mathcal{M}_{t-1}$, there exists a constant $C_\pi > 0$, such that $\left|\frac{\pi^{\mathrm{e}}(a|x)}{\pi_t(a|x, \Omega_{t-1})}\right| \le C_\pi$.*

Assumption 1 equivalently means that $\pi_t(a|x) > 0$ for all $a \in \mathcal{A}$ and $x \in \mathcal{X}$.

**Assumption 2.** *For all $x \in \mathcal{X}$ and $a \in \mathcal{A}$, $\pi_t(a|x, \Omega_{t-1}) - \tilde{\pi}(a|x) \xrightarrow{\mathrm{P}} 0$, where $\tilde{\pi} : \mathcal{A} \times \mathcal{X} \to (0, 1)$.*

**Assumption 3.** *For all $t \in [T]$ and $a \in \mathcal{A}$, there exists a constant $C_Y > 0$, such that $|Y_t(a)| \leq C_Y$.*

Although the reader may feel that Assumption 1–2 and the SUTVA 1 are strong, we adopt it as a simple and basic case for the application, in order to introduce adaptive-fitting and the ADR estimator. We can extend the proposed method for different cases, such as when the data has the structure of batches or when the average of logging policy converges (Kato, 2021).

## 2.2 Notations

We denote $\mathbb{E}[Y_t(a)|x]$ and $\mathrm{Var}(Y_t(a)|x)$ as $f^*(a, x)$ and $v^*(a, x)$, respectively. Let $\hat{f}_t(a, x)$ be an estimator of $f^*(a, x)$ constructed from $\Omega_t$. Let $\mathcal{N}(\mu, \mathrm{var})$ be the normal distribution with the mean $\mu$ and the variance var. For a random variable $Z$ with density $p(z)$ and function $\mu$, let $\|\mu(Z)\|_2 = \left( \int |\mu(z)|^2 p(z) dz \right)^{1/2}$ be the $L^2$-norm.

# 3 Preliminaries of OPVE

## 3.1 Existing estimators

For estimating $R(\pi^{\mathrm{e}})$ from dependent samples, existing studies propose various estimators. An adaptive version of the IPW estimator is defined as $R_T^{\mathrm{IPW}}(\pi^{\mathrm{e}}) = \frac{1}{T} \sum_{t=1}^{T} \sum_{a=1}^{K} \frac{\pi^{\mathrm{e}}(A_t|X_t)\mathbb{1}[A_t=a]Y_t}{\pi_{t-1}(A_t|X_t, \Omega_{t-1})}$. If the model specification is correct, the direct method (DM) estimator $R_T^{\mathrm{DM}}(\pi^{\mathrm{e}}) = \frac{1}{T} \sum_{t=1}^{T} \sum_{a=1}^{K} \pi^{\mathrm{e}}(a|x)\hat{f}_T(a|X_t)$ is known to be consistent to $R(\pi^{\mathrm{e}})$. As an adaptive version of the AIPW estimator, van der Laan (2008) proposed an estimator $\widehat{R}_T^{\mathrm{AIPW}}(\pi^{\mathrm{e}})$ defined as

$$\frac{1}{T} \sum_{t=1}^{T} \sum_{a=1}^{K} \left\{ \frac{\pi^{\mathrm{e}}(a|X_t)\mathbb{1}[A_t=a]\left(Y_t - \hat{f}_{t-1}(a, X_t)\right)}{\pi_{t-1}(a|X_t, \Omega_{t-1})} + \pi^{\mathrm{e}}(a|X_t)\hat{f}_{t-1}(a, X_t) \right\}.$$

Using the martingale property, van der Laan (2008) showed asymptotic normality under Assumption 2.

## 3.2 Asymptotic efficiency

We are often interested in the asymptotic efficiency of estimators. The lower bound of the asymptotic variance is defined for an estimator under some posited models of the DGP described in Section 2. As with the Cramér-Rao lower bound for the parametric model, we can also define the lower bound for the non- or semiparametric model (Bickel et al., 1998). The semiparametric lower bound of the DGP under $p_1(a|x) = \cdots = p_T(a|x) = p(a|x)$ is given as follows (Hahn, 1998; Narita et al., 2019):

$$\Psi(\pi^{\mathrm{e}}, p) = \mathbb{E}\left[ \sum_{a=1}^{K} \frac{\left(\pi^{\mathrm{e}}(a|X_t)\right)^2 v^*(a, X_t)}{p(a|X_t)} + \left( \sum_{a=1}^{K} \pi^{\mathrm{e}}(a|X_t)f^*(a, X_t) - R(\pi^{\mathrm{e}}) \right)^2 \right].$$

The asymptotic variance of the asymptotic distribution is also known as the asymptotic mean squared error (MSE); that is, an OPVE estimator $\hat{R}_T$ achieving the semiparametric lower bound also minimizes the MSE to the true value $R(\pi^{\mathrm{e}})$, $\mathbb{E}\left[ \left( \hat{R}_T - R(\pi^{\mathrm{e}}) \right)^2 \right]$, not just obtaining a tight confidence interval.

## 3.3 Related work

There are various studies related to OPVE, including ATE estimation, under the assumption that samples are i.i.d. (Hahn, 1998; Hirano et al., 2003; Dudík et al., 2011; Wang et al., 2017b; Narita et al., 2019; Bibaut et al., 2019; Oberst & Sontag, 2019). There are also several studies extending these methods to OPVE from dependent samples (van der Laan, 2008; van der Laan & Lendle, 2014; Luedtke & van der Laan, 2016, 2018; Hadad et al., 2021; Kato et al., 2020; Zhang et al., 2020, 2021; Zhan et al., 2021; Bibaut et al., 2021).

The AIPW estimator for dependent samples are proposed by van der Laan (2008). van der Laan & Lendle (2014) and Luedtke & van der Laan (2016) proposed an OPVE method without the convergence of the logging policies by using batches and standardization, respectively. Hadad et al. (2021) and Zhan et al. (2021) extended the method of Luedtke & van der Laan (2016) and proposed using an evaluation weight to stabilize the estimator and to evaluate regret minimization algorithms, where the logging policy converges to 0 for sub-optimal arms. The former shares a similar motivation with weight clipping, or shrinkage, when i.i.d. samples are given (Bembom & van der Laan, 2008; Bottou et al., 2013; Wang et al., 2017a; Su et al., 2019, 2020). Howard et al. (2021) and Kato et al. (2020) showed a non-asymptotic confidence interval of the AIPW estimator. Zhang et al. (2020) proposed an estimator of a linear regression model for samples with batches. Estimators proposed by van der Laan (2008), Luedtke & van der Laan (2016), Hadad et al. (2021), Kato et al. (2020), and Zhang et al. (2020) require the true logging policy, unlike our ADR estimator.

A semiparametric estimator usually requires the Donsker condition for its asymptotic normality (Bickel et al., 1998). For semiparametric inference without the Donsker condition, sample-splitting is a typical approach (Klaassen, 1987; Zheng & van der Laan, 2011; Chernozhukov et al., 2018). Chernozhukov et al. (2018) referred to sample-splitting as *cross-fitting* and the semiparametric inference using cross-fitting as *double-debiased machine learning* (DML). The DML has been extended for several settings (Chiang et al., 2019; Chang, 2020; Jung et al., 2021). For off-policy evaluation of reinforcement learning from dependent samples, Kallus & Uehara (2019) proposed a mixing-based sample-splitting, which requires a different assumption from martingales.

In the recent literature, there has been a growing interest in the evaluation of regret minimization algorithms, where the logging policy $\pi_t(a|x)$ converges to 0 for sub-optimal arms $a \in \mathcal{A}$. In this case, Assumption 1 does not hold. Note that if the logging policy does not converge to 0, we incur a linear regret in the regret minimization problem. Along with stabilizing unstable behavior with the logging policy, Hadad et al. (2021) showed that their proposed method can estimate the policy value of regret minimization algorithms with expected regret close to $\log T$ order. Because the method of Hadad et al. (2021) cannot deal with covariates, Zhan et al. (2021) extended the method in this direction. Their method can evaluate algorithms with the expected regret close to $\sqrt{T}$ order. Unlike those studies, our proposed ADR estimator may not be applicable to this setting. However, we do not consider this to be a practical limitation. The evaluation of regret minimization algorithm is a very important problem, but in industry, uniformly random arm selection and regret minimization algorithm are often used together for sanity checks, in which Assumption 1 is satisfied. It is also believed that using such a uniformly random arm selection will stabilize the logging policy and improve the empirical performance of OPVE. In our paper, we instead focus more on the instability caused by the time-varying logging policy.

## 4 The ADR estimator

For OPVE with dependent samples, this paper proposes the ADR estimator $\widehat{R}_T^{\mathrm{ADR}}(\pi^{\mathrm{e}})$ defined as

$$\frac{1}{T} \sum_{t=1}^{T} \sum_{a=1}^{K} \left\{ \frac{\pi^{\mathrm{e}}(a|X_t) \mathbb{1}[A_t = a] \left( Y_t - \hat{f}_{t-1}(a, X_t) \right)}{\hat{g}_{t-1}(a|X_t)} + \pi^{\mathrm{e}}(a|X_t) \hat{f}_{t-1}(a, X_t) \right\},$$

where $\hat{g}_{t-1}$ is an estimator of $\pi_{t-1}$, constructed only from $\Omega_{t-1}$. We can use standard regression methods for constructing $\hat{f}_{t-1}$ and $\hat{g}_{t-1}$ if they satisfy the following assumptions.

**Assumption 4.** *For $p, q > 0$ such that $p + q = 1/2$, $\|\hat{g}_{t-1}(a|X_t) - \pi_{t-1}(a|X_t, \Omega_{t-1})\|_2 = \mathrm{o}_p(t^{-p})$, and $\|\hat{f}_{t-1}(a, X_t) - f^*(a, X_t)\|_2 = \mathrm{o}_p(t^{-q})$, where the expectation of the norm is taken over $X_t$.*

**Assumption 5.** *There exists a constant $C_f$ such that $|\hat{f}_{t-1}(a, x)| \leq C_f \ \forall a \in \mathcal{A}, x \in \mathcal{X}$.*

**Assumption 6.** *There exist a constant $C_g$ such that $0 < \left| \frac{\pi^{\mathrm{e}}(a|x)}{\hat{g}_{t-1}(a|x)} \right| \leq C_g \ \forall a \in \mathcal{A}, x \in \mathcal{X}$.*

Assumption 4 requires convergence rates standard in regression estimators. For instance, we can apply nonparametric estimators proposed in MAB problems (Yang & Zhu, 2002; Qian & Yang, 2016). Under the assumptions, we show the asymptotic normality of the ADR estimator.

**Theorem 1** (Asymptotic distribution of an ADR estimator). *Under Assumptions 1–6,*

$$\sqrt{T} \left( \widehat{R}_T^{\mathrm{ADR}}(\pi^{\mathrm{e}}) - R(\pi^{\mathrm{e}}) \right) \xrightarrow{d} \mathcal{N} \left( 0, \Psi(\pi^{\mathrm{e}}, \tilde{\pi}) \right).$$

Let us put forward the following assumption.

**Assumption 7.** *For all $x \in \mathcal{X}$ and $a \in \mathcal{A}$, $\hat{f}_{t-1}(a,x) - f^*(a,x) \xrightarrow{\mathrm{p}} 0$.*

The proof is shown in Appendix B, which uses the following proposition from Kato et al. (2020).

**Proposition 1** (Asymptotic distribution of an AIPW estimator (Corollary 1, Kato et al. (2020)).)**.** *Under Assumptions 1–3, 5, and 7, $\sqrt{T}\left(\widehat{R}_T^{\mathrm{AIPW}}(\pi^{\mathrm{e}}) - R(\pi^{\mathrm{e}})\right) \xrightarrow{d} \mathcal{N}\left(0, \Psi(\pi^{\mathrm{e}}, \tilde{\pi})\right)$.*

**Remark 3** (Consistency and double robustness)**.** *The ADR estimator has double robustness, as with standard DR estimators; that is, if either $\hat{f}$ or $\hat{g}$ is consistent, the ADR estimator is also consistent.*

**Theorem 2** (Consistency)**.** *Under Assumptions 1–3 and 5–6, if either $\hat{f}_{t-1}$ or $\hat{g}_{t-1}$ is consistent, $\widehat{R}_T^{\mathrm{ADR}} \xrightarrow{\mathrm{p}} R(\pi^{\mathrm{e}})$.*

We can prove this by the law of large numbers for martingales (Proposition 3 in Appendix A).

**Donsker condition.** The main reason for using step-wise estimators $\{\hat{f}_{t-1}\}_{t=1}^T$ and $\{\hat{g}_{t-1}\}_{t=1}^T$ is to regard them as constants in the expectation conditioned on $\Omega_{t-1}$. The motivation is shared with DML. For asymptotic normality shown in Theorem 1, we do not impose the Donsker condition on the nuisance estimators, $\hat{f}_{t-1}$ and $\hat{g}_{t-1}$, but only require the convergence rate conditions. We call this sample-splitting method, using $\hat{f}_{t-1}$ and $\hat{g}_{t-1}$, *adaptive-fitting* and discuss again in Section 5. In the MAB problem, convergence rate conditions in nonparametric regression, such as a nearest-neighbor regression (Yang & Zhu, 2002), Nadaraya-Watson regression (Qian & Yang, 2016), random forest (Féraud et al., 2016), kernelized linear models (Chowdhury & Gopalan, 2017), and neural networks (Dongruo Zhou, 2020), have been shown. Note that we need to slightly modify the results for each situation because they are influenced by time-series behavior of logging probabilities.

**Convergence rate of the logging policy.** In the main theorem, we do not explicitly describe the convergence rate of the logging policy. However, from Assumption 4, which requires $\|\hat{g}_{t-1}(a|X_t) - \pi_{t-1}(a|X_t, \Omega_{t-1})\|_2 = \mathrm{o}_p(t^{-p})$, $\|\tilde{\pi}(a|x) - \pi_{t-1}(a|X_t, \Omega_{t-1})\|_2 = \mathrm{o}_p(t^{-p})$ is also required.

**Theoretical comparison between AIPW and ADR estimators.** There are two major differences between AIPW and ADR estimators. First, the AIPW estimator requires *a priori* knowledge of the true logging policy, but the ADR estimator does not. Another main difference between the two is the convergence rate of nuisance estimators $\hat{g}_{t-1}$ and $\hat{f}_{t-1}$. For the AIPW estimator, only the uniform convergence in probability is required, but the ADR estimator requires specific convergence rates on $\hat{g}_{t-1}$ and $\hat{f}_{t-1}$. This difference comes from unbiasedness. The AIPW estimator is unbiased; therefore, the convergence of the asymptotic variance is essential, where it converges with $\mathrm{o}_p(1)$ if $\hat{f}$ and $\pi_t$ is $\mathrm{o}_p(1)$. Thus, the AIPW estimator does not require specific convergence rates. On the other hand, the ADR estimator requires the asymptotic bias term to vanish in a specific order. For this purpose, it imposes specific convergence rates on the nuisance estimators. From another perspective, a standard DML for i.i.d. samples and Theorem 1 require $\|\hat{g}_{t-1}(a|X_t) - \pi_{t-1}(a|X_t, \Omega_{t-1})\|_2 = \mathrm{o}_p(t^{-p})$, $\|\hat{f}_{t-1}(a, X_t) - f^*(a, X_t)\|_2 = \mathrm{o}_p(t^{-q})$, and $p + q = 1/2$.

**Asymptotic efficiencies.** As shown in Theorem 1 and Proposition 1, ADR and AIPW estimators achieve the semiparametric lower bound under the DGP defined in Section 2 with $p_1(a|x) = \cdots = p_T(a|x) = p(a|x)$. On the other hand, the asymptotic variance of the IPW estimator using the true logging policy $\pi_t$ is larger than the lower bound (Hirano et al., 2003; van der Laan, 2008; Kato et al., 2020). Although it is known that the IPW estimator using the estimated logging policy can achieve the lower bound under some conditions with i.i.d. samples (Hirano et al., 2003), the asymptotic property under dependent samples is still unknown.

## 5 Adaptive-fitting: DML for dependent samples

We generalize the method used to derive the asymptotic normality of the ADR estimator as a variant of DML. Let us define the parameter of interest $\theta_0$ that satisfies $\mathbb{E}[\psi(W_t; \theta_0, \eta_0)] = 0$, where $\{W_t\}_{t=1}^T$ are observations, $\eta_0$ is a nuisance parameter, and $\psi$ is a score function. We consider obtaining an asymptotic normal estimator of $\theta_0$ when using complex and data-adaptive regression methods, such as random forests, neural networks, and Lasso, to estimate the nuisance parameter $\eta_0$. Under such a situation, the Donsker condition does not hold, in general. Sample-splitting is a typical approach to

control the complexities of semiparametric inference without the Donsker condition (Klaassen, 1987; Zheng & van der Laan, 2011; Chernozhukov et al., 2018). In DML of Chernozhukov et al. (2018), the dataset with i.i.d. samples $\{W_t\}_{t=1}^T$ is separated into several subgroups. Then, a semiparametric estimator for each subgroup is constructed, but the nuisance estimators are constructed from the other subgroups. Here, the nuisance estimators are independent in the expectation conditioned on the other subgroups. Thus, the complexities are controlled without the Donsker condition, and standard nonparametric convergence rate conditions on nuisance estimators suffice to show the asymptotic normality of the semiparametric estimator. Chernozhukov et al. (2018) called the method cross-fitting.

However, cross-fitting of Chernozhukov et al. (2018) cannot be applied when samples $\{W_t\}_{t=1}^T$ are dependent. Therefore, the ADR estimator uses step-wise nuisance estimators based on past observations $\Omega_{t-1}$. This construction is inspired by van der Laan & Lendle (2014), which proposed a sample-splitting for batched dependent samples. The nuisance estimators are independent in the expectation over the $t$-th samples conditioned on $\Omega_{t-1}$. Thus, we can regard our sample-splitting as another approach

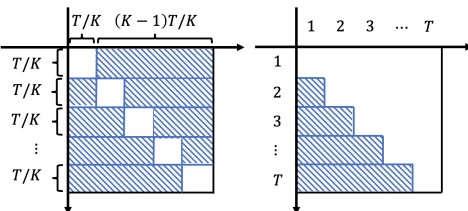

Figure 1: The difference between cross-fitting (left) and adaptive-fitting (right).

for DML. We call this step-wise construction *adaptive-fitting*, in contrast to cross-fitting. We briefly summarize the procedure as follows: (i) at each step $t = 1, 2, \ldots, T$, we estimate $\eta_0$ only using $\{W_s\}_{s=1}^{t-1}$ and denote the estimators as $\{\eta_{t-1}\}_{t=1}^T$; (ii) then, we substitute $W_t$ and $\eta_{t-1}$ into $\psi$ and obtain an estimator $\hat{\theta}_T$ of $\theta_0$ by solving $\frac{1}{T}\sum_{t=1}^T \psi(W_t, \hat{\theta}_T, \eta_{t-1}) = 0$. Under this construction, if (a) an estimator $\check{\theta}_T$ obtained by solving $\frac{1}{T}\sum_{t=1}^T \psi(W_t, \check{\theta}_T, \eta_0) = 0$ has the asymptotically normal distribution and (b) the asymptotic bias decays with the rate faster than $o_p(-1/\sqrt{t})$, we can obtain the asymptotically normal estimator of $\theta_0$. In Figure 1, we illustrate the difference between cross-fitting (left) and adaptive-fitting (right). The y-axis represents samples used for constructing the OPVE estimator with nuisance estimators using samples represented by the x-axis. In the left graph, for $T/K$ samples, we calculate the sample average of the component including nuisance estimators based on the other $(K-1)T/K$ samples. In the right graph, at period $t$, we use nuisance estimators constructed from samples at $t = 1, 2, \ldots, t-1$. To satisfy (a), we assumed that the convergence of the logging policy (Assumption 2). There are other ways to satisfy this condition. For example, we can assume the existence of batches as van der Laan & Lendle (2014) and Zhang et al. (2020). We briefly introduce the ADR estimator with this case in Appendix C.

**Remark 4** (The EIPW and DM estimator). *To obtain asymptotic normality via DML, we need a doubly robust structure on the estimator to reduce the asymptotic bias caused by the estimation errors of the nuisance parameters. For this reason, even by using DML, we cannot show the asymptotic normality of the IPW estimator, with an estimated logging policy (EIPW estimator), and the DM estimator, which do not have the doubly robust structure. For example, let us define the EPW estimator as $\frac{1}{T}\sum_{t=1}^T \sum_{a=1}^K \frac{\pi^{\mathrm{e}}(a|X_t)\mathbb{1}[A_t=a]Y_t(a)}{\hat{g}_{t-1}(a|X_t)}$. Then, the bias is $\left| \frac{1}{T}\sum_{t=1}^T \sum_{a=1}^K \frac{\pi^{\mathrm{e}}(a|X_t)\mathbb{1}[A_t=a]Y_t(a)}{\hat{g}_{t-1}(a|X_t)} - \frac{1}{T}\sum_{t=1}^T \sum_{a=1}^K \frac{\pi^{\mathrm{e}}(a|X_t)\mathbb{1}[A_t=a]Y_t(a)}{\pi_{t-1}(a|X_t,\Omega_{t-1})} \right|$. Here, because the convergence rate of the estimation error $\|\pi_{t-1}(a|X_t,\Omega_{t-1}) - \hat{g}_{t-1}(a|X_t)\|_2$ will not exceed $O_p(1/\sqrt{T})$, the bias does not vanish with $o_p(1/\sqrt{T})$, which implies that we cannot ignore the bias when deriving the asymptotic normality. Similarly, the DM estimator also suffers from the estimation error $\|f^*(a, X_t) - \hat{f}_{t-1}(a, X_t)\|_2$. On the other hand, when using the DR estimator, the bias depends on $\|\pi_{t-1}(a|X_t,\Omega_{t-1}) - \hat{g}_{t-1}(a|X_t)\|_2 \|f^*(a, X_t) - \hat{f}_{t-1}(a, X_t)\|_2$, which can be $o_p(1/\sqrt{T})$ and ignored to derive the asymptotic normality. Note that when the samples are i.i.d., by using a specific estimator for $\pi_t$, we can show the asymptotic normality for the EIPW estimator. See Hirano et al. (2003) and Hahn et al. (2011) for details.*

## 6 Paradox concerning logging policy

The advantage of the ADR estimator is that, unlike the AIPW estimator, it does not require the true logging policy. However, we find that the ADR estimator often achieves a smaller MSE than the

Table 1: The results of Section 6.1 with sample sizes $T = 250, 500, 750$. We show the RMSE, SD, and coverage ratio of the confidence interval (CR). We highlight in red bold two estimators with the lowest RMSE. Estimators with asymptotic normality are marked with †, and estimators that do not require the true logging policy are marked with ∗.

| | LinUCB policy | | | | | | | | | | | | | | | | | |
|---|---|---|---|---|---|---|---|---|---|---|---|---|---|---|---|---|---|---|
| $T$ | ADR †∗ | | | IPW † | | | AIPW † | | | AW-AIPW † | | | DM ∗ | | | EIPW ∗ | | |
| | RMSE | SD | CR | RMSE | SD | CR | RMSE | SD | CR | RMSE | SD | CR | RMSE | SD | CR | RMSE | SD | CR |
| 250 | **0.061** | 0.004 | 0.93 | 0.205 | 0.298 | 0.89 | 0.158 | 0.129 | 0.94 | 0.179 | 0.180 | 0.64 | 0.073 | 0.006 | 0.17 | 0.097 | 0.012 | 0.81 |
| 500 | **0.039** | 0.002 | 0.97 | 0.102 | 0.029 | 0.89 | 0.078 | 0.012 | 0.92 | 0.076 | 0.011 | 0.61 | 0.049 | 0.003 | 0.19 | 0.124 | 0.014 | 0.34 |
| 750 | **0.033** | 0.001 | 0.95 | 0.122 | 0.081 | 0.92 | 0.088 | 0.041 | 0.95 | 0.095 | 0.050 | 0.72 | 0.038 | 0.002 | 0.21 | 0.134 | 0.013 | 0.13 |
| | LinTS policy | | | | | | | | | | | | | | | | | |
| $T$ | ADR †∗ | | | IPW † | | | AIPW † | | | AW-AIPW † | | | DM ∗ | | | EIPW ∗ | | |
| | RMSE | SD | CR | RMSE | SD | CR | RMSE | SD | CR | RMSE | SD | CR | RMSE | SD | CR | RMSE | SD | CR |
| 250 | **0.060** | 0.004 | 0.93 | 0.121 | 0.025 | 0.85 | 0.110 | 0.026 | 0.88 | 0.126 | 0.038 | 0.63 | 0.073 | 0.006 | 0.14 | 0.088 | 0.010 | 0.82 |
| 500 | **0.038** | 0.001 | 0.96 | 0.087 | 0.010 | 0.94 | 0.065 | 0.006 | 0.97 | 0.070 | 0.008 | 0.63 | 0.046 | 0.002 | 0.10 | 0.103 | 0.010 | 0.46 |
| 750 | **0.030** | 0.001 | 0.97 | 0.059 | 0.006 | 0.96 | 0.060 | 0.009 | 0.94 | 0.062 | 0.009 | 0.68 | 0.038 | 0.002 | 0.17 | 0.116 | 0.011 | 0.22 |

Figure 2: This figure illustrates the error distributions of estimators for OPVE from dependent samples generated with the LinUcB(left) and LinTS(right) with the sample size 750. We smoothed the error distributions using kernel density estimation. Estimators with asymptotic normality are marked with †, and estimators that do not require a true logging policy are marked with ∗.

AIPW estimator, which requires the true logging policy. We refer to this phenomenon as a paradox concerning logging policy.

## 6.1 Instability of the AIPW estimator

Hadad et al. (2021) pointed out that the AIPW estimator tends to be unstable, and the statistical inference becomes difficult when the nuisance parameter $\pi_{t-1}$ is close to or converges to $0$. To prevent this instability and make statistical inference possible, Hadad et al. (2021) and Zhan et al. (2021) proposed the Adaptively Weighted AIPW (AW-AIPW) estimator by adding evaluation weights. The purpose of these methods is twofold: to stabilize the estimator and expand the algorithm class whose policy value can be evaluated. In our paper, we focus more on the former stabilization problem.

The ADR estimator replaces the true logging policy $\pi_{t-1}$ with its estimator. We find that by replacing the true logging policy with its estimator, we can control the instability caused from $\pi_{t-1}$ by constructing well-formed $\hat{g}_{t-1}$. For example, if we know that the range of $\tilde{\pi}$ is $(\varepsilon, 1 - \varepsilon)$ for $0 < \varepsilon < 1 - \varepsilon < 1$, we can add the clipping technique when estimating $\hat{g}_{t-1}$. For example, in early periods, we can set $\hat{g}_{t-1}$ as $0.5$. Such techniques is greatly beneficial. Note that if the range of $\tilde{\pi}$ is truly $(\varepsilon, 1 - \varepsilon)$, the clipping of $\hat{g}_{t-1}$ does not cause clipping bias; that is, the estimator correctly converges to the asymptotic distribution shown in Theorem 1.

The main difference between the AW-AIPW estimator and the ADR estimator is that the former requires information of the true logging policy $\pi_{t-1}$, while the latter does not by replacing it with its estimator. Besides, in addition to the stabilization, the AW-AIPW has another purpose: to evaluate the regret minimization algorithms, in which the logging policy converges to $0$, but our proposed method does not solve the problem because we put an assumption on the bounded support (Assumption 1). Our finding also shares motivation with weight clipping for i.i.d. samples, such as Bottou et al. (2013). We can regard the AW-AIPW estimator as a variant of this method, where the weight clipping can decay as $\pi_t$ converges, and be ignored asymptotically. However, our interest is in asymptotic normality from dependent samples without the true logging policy, which is not discussed in the literature.

## 6.2 Simulation studies for OPVE estimators

For reasons given in the previous section, the ADR estimator may empirically perform better than the AIPW estimator because the estimator $\hat{g}_{t-1}$ stabilizes the ADR estimator by absorbing the instability of $\pi_{t-1}$. We empirically show this paradox using a synthetic dataset. We generate an artificial pair of covariate and potential outcome, $(X_t, Y_t(1), Y_t(2), Y_t(3))$. The covariate $X_t$ is a 10 dimensional vector generated from the standard normal distribution. For $a \in \{1, 2, 3\}$, $Y_t(a) = 1$ if $a$ is chosen with probability $q(a|x) = \frac{\exp(g(a,x))}{\sum_{a'=1}^{3} \exp(g(a',x))}$, where $g(1,x) = \sum_{d=1}^{10} X_{t,d}$, $g(2,x) = \sum_{d=1}^{10} W_d X_{t,d}^2$, $g(3,x) = \sum_{d=1}^{10} W_d |X_{t,d}|$, and $W_d$ is uniform randomly chosen from $\{-1, 1\}$. We generate three datasets, $\mathcal{S}_{T_{(1)}}^{(1)}$, $\mathcal{S}_{T_{(2)}}^{(2)}$, and $\mathcal{S}_{T_{(3)}}^{(3)}$, where $\mathcal{S}_{T_{(m)}}^{(m)} = \{(X_t^{(m)}, Y_t^{(m)}(1), Y_t^{(m)}(2), Y_t^{(m)}(3))\}_{t=1}^{T_{(m)}}$. First, we train an evaluation function $\pi^e$ by solving a prediction problem between $X_t^{(1)}$ and $Y_t^{(1)}(1), Y_t^{(1)}(2), Y_t^{(1)}(3)$ using $\mathcal{S}_{T_{(1)}}^{(1)}$. Then, we regard the evaluation function $\pi^e$ as a logging policy and apply it on the independent dataset $\mathcal{S}_{T_{(2)}}^{(2)}$ to obtain $\{(X_t', A_t', Y_t')\}_{t=1}^{T_{(2)}}$, where $A_t'$ is an action chosen from the evaluation function and $Y_t^{(m)} = \sum_{a=1}^{3} \mathbb{1}[A_t^{(m)} = a] Y_t^{(m)}(a)$. Then, we estimate the true value $R(\pi^e)$ as $\frac{1}{T_{(2)}} \sum_{t=1}^{T_{(2)}} Y_t^{(m)}$. Next, using the datasets $\mathcal{S}_{T_{(3)}}^{(3)}$ and an MAB algorithm, we generate a bandit dataset $\mathcal{S} = \{(X_t, A_t, Y_t)\}_{t=1}^{T_{(3)}}$. For $\mathcal{S}$, we apply the ADR estimator, IPW estimator with the true logging policy (IPW), AIPW estimator (AIPW), AW-AIPW estimator (AW-AIPW) IPW estimator with estimated logging policy (EIPW), and DM estimator (DM). For the AW-AIPW estimator, we use the `StableVar` weight, proposed in Zhan et al. (2021). For estimating $f^*$ and the logging policy, we use the kernelized Ridge least squares and logistic regression, respectively. We use a Gaussian kernel, and both hyper-parameters of the Ridge and kernel are chosen from $\{0.01, 0.1, 1\}$. We define the estimation error as $R(\pi^e) - \widehat{R}(\pi^e)$. We conduct ten experiments by changing sample sizes and MAB algorithms. For the sample size $T_{(3)}$, we use 100, 250, 500, 750, and 10,000 with the LinUCB and LinTS algorithms. For the sample sizes $T_{(1)}$ and $T_{(2)}$, we use 1,000 and 100,000 respectively. We conduct 100 trials to obtain the root MSEs (RMSEs), the standard deviations of MSEs (SDs), and the coverage ratios (CRs) of the 95% confidence interval (percentage that the confidence interval covers the true value). The results with $T = 250, 500, 750$ are shown in Table 1 and those with $T = 750$ are shown in Figure 2. The other results are shown in Appendix D. In all cases, the ADR estimator performs well compared to other methods. Although the AIPW estimator has asymptotic normality, the performance is not comparable with the ADR estimator. Although the AW-AIPW estimator enables us to evaluate regret minimization algorithms, we cannot observe that the AW-AIPW estimator improves the empirical performance of the AIPW estimator in our setting. The EIPW estimator suffers from the dependency problem.

**Remark 5** (Paradox for i.i.d. samples). *This paradox is similar to the well-known property that the IPW estimator using an estimated propensity score shows a smaller asymptotic variance than the IPW using the true one (Hirano et al., 2003; Henmi & Eguchi, 2004; Henmi et al., 2007). However, as discussed above, we consider this to be a different phenomenon. In these studies, the paradox is mainly explained by differences in the asymptotic variance between IPW estimators with the true and estimated propensity score. On the other hand, for our case, AIPW and ADR estimators have the same asymptotic variance, unlike IPW-type estimators. Therefore, we cannot resolve the paradox by the aforementioned explanations. The instability of the AIPW estimator is consistent with the empirical findings of Hadad et al. (2021), but, in our paper, we suggest using the ADR estimator as another solution.*

## 7 Experiments

Following Dudík et al. (2011), we evaluate the estimators using classification datasets by transforming them into contextual bandit data. From the LIBSVM repository (Chang & Lin, 2011), we use the `mnist`, `satimage`, `sensorless`, and `connect-4` datasets [2]. The dataset description is in Table 3 of Appendix D. We construct a logging policy as $\pi_t(a|x) = \alpha \pi_t^m(a|x) + (1 - \alpha)\frac{0.1}{K}$, where $\alpha \in (0, 1)$ is a constant, and $\pi_t^m(a|x)$ is a policy such that $\pi_t^m(a|x) = 1$ for an action $a \in \mathcal{A}$ and $\pi_t^m(a'|x) = 0$ for the other actions ($a' \neq a$). The policy $\pi_t^m(a|x)$ is determined by logistic regression and MAB

---

[2] https://www.csie.ntu.edu.tw/~cjlin/libsvmtools/datasets/.

Table 2: The results of benchmark datasets with the LinUCB policy. We highlight in red bold the estimator with the lowest RMSE and highlight in under line the estimator with the lowest RMSE among estimators that do not use the true logging policy. Estimators with asymptotic normality are marked with †, and estimators that do not require the true logging policy are marked with ∗.

| mnist | ADR †∗ | | IPW † | | AIPW † | | DM ∗ | | EIPW ∗ | |
|---|---|---|---|---|---|---|---|---|---|---|
| $\alpha$ | RMSE | SD | RMSE | SD | RMSE | SD | RMSE | SD | RMSE | SD |
| 0.7 | **0.046** | 0.002 | 0.100 | 0.011 | 0.162 | 0.027 | 0.232 | 0.013 | 0.148 | 0.014 |
| 0.4 | **0.028** | 0.001 | 0.068 | 0.005 | 0.112 | 0.009 | 0.249 | 0.010 | 0.080 | 0.004 |
| 0.1 | 0.086 | 0.006 | **0.078** | 0.006 | 0.085 | 0.008 | 0.299 | 0.024 | 0.091 | 0.008 |

| satimage | ADR †∗ | | IPW † | | AIPW † | | DM ∗ | | EIPW ∗ | |
|---|---|---|---|---|---|---|---|---|---|---|
| $\alpha$ | RMSE | SD | RMSE | SD | RMSE | SD | RMSE | SD | RMSE | SD |
| 0.7 | **0.013** | 0.000 | 0.098 | 0.011 | 0.060 | 0.004 | 0.037 | 0.001 | 0.056 | 0.002 |
| 0.4 | **0.022** | 0.000 | 0.078 | 0.008 | 0.019 | 0.000 | 0.043 | 0.001 | 0.060 | 0.002 |
| 0.1 | **0.029** | 0.001 | 0.078 | 0.005 | 0.061 | 0.008 | 0.041 | 0.002 | 0.041 | 0.002 |

algorithms. For MAB algorithms, we use upper confidence bound and Thompson sampling with a linear model, which are denoted as LinUCB (Chu et al., 2011) and LinTS (Agrawal & Goyal, 2013). The evaluation function is fixed at $\pi^{\mathrm{e}}(a|x) = 0.9\pi^d(a|x) + \frac{0.1}{K}$, where $\pi_d$ is a prediction of a logistic regression.

We focus on the ADR estimator and compare it to the IPW, EIPW, AIPW, and DM estimators, as done in Section 6.2. We omit the AW-AIPW estimator, as we found that the performance depends on the choice of the evaluation weight (see Section 6). Additionally, the paper does not consider or discuss situations where there exist covariates. We compare those estimators using the benchmark datasets generated through the process explained above. For $\alpha \in \{0.7, 0.4, 0.1\}$ and the sample sizes 800, 1,000, and 1,200, we calculate the RMSEs and the SDs over 10 trials. The results of `mnist` and `satimage` with the LinUCB policy and $T = 1,000$ are shown in Tables 2. The full results, including the results with the LinTS policy and i.i.d. samples, are shown in Appendix D. As discussed in Section 6, the ADR estimator performs well. Although the asymptotic distributions of AIPW and ADR estimators are the same, the AIPW estimator shows poorer performance. As Section 6.2, we consider this to be because the estimator of $\pi_t$ is absorbing the instability of $\pi_t$.

Note that the ADR, IPW, and AIPW estimators are asymptotically normal, but the IPW and APIW estimators are not feasible when the true logging policy is not given. To the best of our knowledge, asymptotic normality of the EIPW and DM estimators has not been shown when samples are dependent, and the proof is non-trivial owing to the dependency and the Donsker condition.

# 8 Conclusion

DR-type estimators are crucial in causal inference because they do not assume *a priori* knowledge of the true logging policy, and they asymptotically follow a normal distribution under standard convergence rate conditions of nuisance estimators. However, existing studies have rarely discussed DR-type estimators when samples are dependent. We derived the ADR estimator and proposed adaptive-fitting as a variant of DML to obtain asymptotic normality. In experiments, we found a paradox that the ADR estimator tends to be more stable than the AIPW estimator and conjectured that this is because the ADR estimator absorbs the instability of the true logging policy $\pi_t$.

In OPVE with dependent samples, to obtain asymptotic normality, we need to put some assumptions on the behavior of $\pi_t$, such as Assumption 2 and a decay rate condition assumed in Hadad et al. (2021) and Zhan et al. (2021). These assumptions can be broken if the time-series is very complicated. However, we note that this is a limitation of the entire field. The asymptotic normality and double robustness of OPVE estimators are necessary theoretical properties to avoid deriving false causality. Since causal inference is often used in applications related closely to public policy, we consider understanding these limitations critical.

# Acknowledgments

We thank Masatoshi Uehara and Kaito Ariu for insightful comments and discussion.

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
