## A  Mathematical preliminaries

**Proposition 2** ($L^r$ Convergence Theorem, Loeve (1977)). *Let $0 < r < \infty$, suppose that $\mathbb{E}\big[|a_n|^r\big] < \infty$ for all $n$ and that $a_n \xrightarrow{\text{P}} a$ as $n \to \infty$. The following are equivalent:*

*(i) $a_n \to a$ in $L^r$ as $n \to \infty$;*

*(ii) $\mathbb{E}\big[|a_n|^r\big] \to \mathbb{E}\big[|a|^r\big] < \infty$ as $n \to \infty$;*

*(iii) $\big\{|a_n|^r, n \geq 1\big\}$ is uniformly integrable.*

**Proposition 3** (Weak Law of Large Numbers for Martingale, Hall et al. (2014)). *Let $\{S_n = \sum_{i=1}^{n} X_i, \mathcal{H}_t, t \geq 1\}$ be a martingale and $\{b_n\}$ a sequence of positive constants with $b_n \to \infty$ as $n \to \infty$. Then, writing $X_{ni} = X_i \mathbb{1}[|X_i| \leq b_n]$, $1 \leq i \leq n$, we have that $b_n^{-1} S_n \xrightarrow{\text{P}} 0$ as $n \to \infty$ if*

**(i)** $\sum_{i=1}^{n} P(|X_i| > b_n) \to 0$;

**(ii)** $b_n^{-1} \sum_{i=1}^{n} \mathbb{E}[X_{ni}|\mathcal{H}_{t-1}] \xrightarrow{\text{P}} 0$, and;

**(iii)** $b_n^{-2} \sum_{i=1}^{n} \big\{\mathbb{E}[X_{ni}^2] - \mathbb{E}\big[\mathbb{E}\big[X_{ni}|\mathcal{H}_{t-1}\big]\big]^2\big\} \to 0$.

**Remark 6.** *The weak law of large numbers for martingale holds when the random variable is bounded by a constant.*

## B  Proof of Theorem 1

*Proof of Theorem 1.* We show asymptotic normality of

$$\widehat{R}_T^{\text{ADR}}(\pi^{\text{e}}) = \frac{1}{T} \sum_{t=1}^{T} \left\{ \phi_1(X_t, A_t, Y_t; \hat{g}_{t-1}, \hat{f}_{t-1}) + \phi_2(X_t; \hat{f}_{t-1}) \right\},$$

where

$$\phi_1(X_t, A_t, Y_t; g, f) = \sum_{a=1}^{K} \frac{\pi^{\text{e}}(a|X_t) \mathbb{1}[A_t = a] (Y_t - f(a, X_t))}{g(a|X_t)}$$

$$\phi_2(X_t; f) = \sum_{a=1}^{K} \pi^{\text{e}}(a|X_t) f(a, X_t).$$

Let us define an AIPW estimator with $\hat{f} = f^*$ as

$$\widehat{R}^*(\pi^{\text{e}}) = \frac{1}{T} \sum_{t=1}^{T} \left\{ \phi_1(X_t, A_t, Y_t; \pi_{t-1}, f^*) + \phi_2(X_t; f^*) \right\}.$$

We decompose $\sqrt{T}\left(R^{\text{ADRE}}(\pi^{\text{e}}) - R(\pi^{\text{e}})\right)$ as

$$\sqrt{T}\left(R^{\text{ADRE}}(\pi^{\text{e}}) - R(\pi^{\text{e}})\right) = \sqrt{T}\left(\widehat{R}_T^{\text{ADR}}(\pi^{\text{e}}) - \widehat{R}^*(\pi^{\text{e}}) + \widehat{R}^*(\pi^{\text{e}}) - R(\pi^{\text{e}})\right).$$

From Proposition 1 of Kato et al. (2020) and Assumption 1 and 3, because $\sqrt{T}\left(\widehat{R}^*(\pi^{\text{e}}) - R(\pi^{\text{e}})\right)$ follows asymptotic normal distribution, we want to show

$$\widehat{R}_T^{\text{ADR}}(\pi^{\text{e}}) - \widehat{R}^*(\pi^{\text{e}}) = o_p(1/\sqrt{T}).$$

Here, we have

$$\widehat{R}_T^{\mathrm{ADR}}(\pi^{\mathrm{e}}) - \widehat{R}^*(\pi^{\mathrm{e}})$$

$$= \frac{1}{T}\sum_{t=1}^{T}\left\{\phi_1(X_t, A_t, Y_t; \hat{g}_{t-1}, \hat{f}_{t-1}) - \phi_1(X_t, A_t, Y_t; \pi_{t-1}, f^*)\right.$$

$$- \mathbb{E}\left[\phi_1(X_t, A_t, Y_t; \hat{g}_{t-1}, \hat{f}_{t-1}) - \phi_1(X_t, A_t, Y_t; \pi_{t-1}, f^*)|\Omega_{t-1}\right]$$

$$\left. + \phi_2(X_t; \hat{f}_{t-1}) - \phi_2(X_t; f^*) - \mathbb{E}\left[\phi_2(X_t; \hat{f}_{t-1}) - \phi_2(X_t; f^*)|\Omega_{t-1}\right]\right\}$$

$$+ \frac{1}{T}\sum_{t=1}^{T}\mathbb{E}\left[\phi_1(X_t, A_t, Y_t; \hat{g}_{t-1}, \hat{f}_{t-1})|\Omega_{t-1}\right] + \frac{1}{T}\sum_{t=1}^{T}\mathbb{E}\left[\phi_2(X_t; \hat{f}_{t-1})|\Omega_{t-1}\right]$$

$$- \frac{1}{T}\sum_{t=1}^{T}\mathbb{E}\left[\phi_1(X_t, A_t, Y_t; \pi_{t-1}, f^*)|\Omega_{t-1}\right] - \frac{1}{T}\sum_{t=1}^{T}\mathbb{E}\left[\phi_2(X_t; f^*)|\Omega_{t-1}\right].$$

In the following parts, we separately show that

$$\sqrt{T}\frac{1}{T}\sum_{t=1}^{T}\left\{\phi_1(X_t, A_t, Y_t; \hat{g}_{t-1}, \hat{f}_{t-1}) - \phi_1(X_t, A_t, Y_t; \pi_{t-1}, f^*)\right. \tag{1}$$

$$- \mathbb{E}\left[\phi_1(X_t, A_t, Y_t; \hat{g}_{t-1}, \hat{f}_{t-1}) - \phi_1(X_t, A_t, Y_t; \pi_{t-1}, f^*)|\Omega_{t-1}\right]$$

$$\left. + \phi_2(X_t; \hat{f}_{t-1}) - \phi_2(X_t; f^*) - \mathbb{E}\left[\phi_2(X_t; \hat{f}_{t-1}) - \phi_2(X_t; f^*)|\Omega_{t-1}\right]\right\}$$

$$= \mathrm{o}_p(1);$$

and

$$\frac{1}{T}\sum_{t=1}^{T}\mathbb{E}\left[\phi_1(X_t, A_t, Y_t; \hat{g}_{t-1}, \hat{f}_{t-1})|\Omega_{t-1}\right] + \frac{1}{T}\sum_{t=1}^{T}\mathbb{E}\left[\phi_2(X_t; \hat{f}_{t-1})|\Omega_{t-1}\right] \tag{2}$$

$$- \frac{1}{T}\sum_{t=1}^{T}\mathbb{E}\left[\phi_1(X_t, A_t, Y_t; \pi_{t-1}, f^*)|\Omega_{t-1}\right] - \frac{1}{T}\sum_{t=1}^{T}\mathbb{E}\left[\phi_2(X_t; f^*)|\Omega_{t-1}\right] = \mathrm{o}_p(1/\sqrt{T}).$$

**Proof of** (1). For any $\varepsilon > 0$, to show that

$$\mathbb{P}\left(\left|\sqrt{T}\frac{1}{T}\sum_{t=1}^{T}\left\{\phi_1(X_t, A_t, Y_t; \hat{g}_{t-1}, \hat{f}_{t-1}) - \phi_1(X_t, A_t, Y_t; \pi_{t-1}, f^*)\right.\right.\right.$$

$$- \mathbb{E}\left[\phi_1(X_t, A_t, Y_t; \hat{g}_{t-1}, \hat{f}_{t-1}) - \phi_1(X_t, A_t, Y_t; \pi_{t-1}, f^*)|\Omega_{t-1}\right]$$

$$\left.\left.\left. + \phi_2(X_t; \hat{f}_{t-1}) - \phi_2(X_t t; f^*) - \mathbb{E}\left[\phi_2(X_t; \hat{f}_{t-1}) - \phi_2(X_t; f^*)|\Omega_{t-1}\right]\right\}\right| > \varepsilon\right)$$

$$\to 0,$$

we show that the mean is $0$ and the variance of the component converges to $0$. Then, from the Chebyshev's inequality, this result yields the statement.

The mean is calculated as

$$
\sqrt{T}\frac{1}{T}\sum_{t=1}^{T}\mathbb{E}\Bigg[\Bigg\{\phi_1(X_t,A_t,Y_t;\hat{g}_{t-1},\hat{f}_{t-1})-\phi_1(X_t,A_t,Y_t;\pi_{t-1},f^*)
$$
$$
-\mathbb{E}\Big[\phi_1(X_t,A_t,Y_t;\hat{g}_{t-1},\hat{f}_{t-1})-\phi_1(X_t,A_t,Y_t;\pi_{t-1},f^*)|\Omega_{t-1}\Big]
$$
$$
+\phi_2(X_t;\hat{f}_{t-1})-\phi_2(X_t;f^*)-\mathbb{E}\Big[\phi_2(X_t;\hat{f}_{t-1})-\phi_2(X_t;f^*)|\Omega_{t-1}\Big]\Bigg\}\Bigg]
$$
$$
=\sqrt{T}\frac{1}{T}\sum_{t=1}^{T}\mathbb{E}\Bigg[\mathbb{E}\Bigg[\Bigg\{\phi_1(X_t,A_t,Y_t;\hat{g}_{t-1},\hat{f}_{t-1})-\phi_1(X_t,A_t,Y_t;\pi_{t-1},f^*)
$$
$$
-\mathbb{E}\Big[\phi_1(X_t,A_t,Y_t;\hat{g}_{t-1},\hat{f}_{t-1})-\phi_1(X_t,A_t,Y_t;\pi_{t-1},f^*)|\Omega_{t-1}\Big]
$$
$$
+\phi_2(X_t;\hat{f}_{t-1})-\phi_2(X_t;f^*)-\mathbb{E}\Big[\phi_2(X_t;\hat{f}_{t-1})-\phi_2(X_t;f^*)|\Omega_{t-1}\Big]\Bigg\}|\Omega_{t-1}\Bigg]\Bigg]
$$
$$
=0
$$

Because the mean is 0, the variance is

$$
\mathrm{Var}\Bigg(\sqrt{T}\frac{1}{T}\sum_{t=1}^{T}\Bigg\{\phi_1(X_t,A_t,Y_t;\hat{g}_{t-1},\hat{f}_{t-1})-\phi_1(X_t,A_t,Y_t;\pi_{t-1},f^*)
$$
$$
-\mathbb{E}\Big[\phi_1(X_t,A_t,Y_t;\hat{g}_{t-1},\hat{f}_{t-1})-\phi_1(X_t,A_t,Y_t;\pi_{t-1},f^*)|\Omega_{t-1}\Big]
$$
$$
+\phi_2(X_t;\hat{f}_{t-1})-\phi_2(X_t;f^*)-\mathbb{E}\Big[\phi_2(X_t;\hat{f}_{t-1})-\phi_2(X_t;f^*)|\Omega_{t-1}\Big]\Bigg\}\Bigg)
$$
$$
=\mathbb{E}\Bigg[\Bigg(\sqrt{T}\frac{1}{T}\sum_{t=1}^{T}\Bigg\{\phi_1(X_t,A_t,Y_t;\hat{g}_{t-1},\hat{f}_{t-1})-\phi_1(X_t,A_t,Y_t;\pi_{t-1},f^*)
$$
$$
-\mathbb{E}\Big[\phi_1(X_t,A_t,Y_t;\hat{g}_{t-1},\hat{f}_{t-1})-\phi_1(X_t,A_t,Y_t;\pi_{t-1},f^*)|\Omega_{t-1}\Big]
$$
$$
+\phi_2(X_t;\hat{f}_{t-1})-\phi_2(X_t;f^*)-\mathbb{E}\Big[\phi_2(X_t;\hat{f}_{t-1})-\phi_2(X_t;f^*)|\Omega_{t-1}\Big]\Bigg\}\Bigg)^2\Bigg]
$$
$$
=\frac{1}{T}\mathbb{E}\Bigg[\Bigg(\sum_{t=1}^{T}\Bigg\{\phi_1(X_t,A_t,Y_t;\hat{g}_{t-1},\hat{f}_{t-1})-\phi_1(X_t,A_t,Y_t;\pi_{t-1},f^*)
$$
$$
-\mathbb{E}\Big[\phi_1(X_t,A_t,Y_t;\hat{g}_{t-1},\hat{f}_{t-1})-\phi_1(X_t,A_t,Y_t;\pi_{t-1},f^*)|\Omega_{t-1}\Big]
$$
$$
+\phi_2(X_t;\hat{f}_{t-1})-\phi_2(X_t;f^*)-\mathbb{E}\Big[\phi_2(X_t;\hat{f}_{t-1})-\phi_2(X_t;f^*)|\Omega_{t-1}\Big]\Bigg\}\Bigg)^2\Bigg].
$$

Therefore, we have

$$
= \frac{1}{T} \sum_{t=1}^{T} \mathbb{E} \Bigg[ \Bigg( \phi_1(X_t, A_t, Y_t; \hat{g}_{t-1}, \hat{f}_{t-1}) - \phi_1(X_t, A_t, Y_t; \pi_{t-1}, f^*)
$$

$$
- \mathbb{E} \Big[ \phi_1(X_t, A_t, Y_t; \hat{g}_{t-1}, \hat{f}_{t-1}) - \phi_1(X_t, A_t, Y_t; \pi_{t-1}, f^*) | \Omega_{t-1} \Big]
$$

$$
+ \phi_2(X_t; \hat{f}_{t-1}) - \phi_2(X_t; f^*) - \mathbb{E} \Big[ \phi_2(X_t; \hat{f}_{t-1}) - \phi_2(X_t; f^*) | \Omega_{t-1} \Big] \Bigg)^2 \Bigg]
$$

$$
+ \frac{2}{T} \sum_{t=1}^{T-1} \sum_{s=t+1}^{T} \mathbb{E} \Bigg[ \Bigg( \phi_1(X_t, A_t, Y_t; \hat{g}_{t-1}, \hat{f}_{t-1}) - \phi_1(X_t, A_t, Y_t; \pi_{t-1}, f^*)
$$

$$
- \mathbb{E} \Big[ \phi_1(X_t, A_t, Y_t; \hat{g}_{t-1}, \hat{f}_{t-1}) - \phi_1(X_t, A_t, Y_t; \pi_{t-1}, f^*) | \Omega_{t-1} \Big]
$$

$$
+ \phi_2(X_t; \hat{f}_{t-1}) - \phi_2(X_t; f^*) - \mathbb{E} \Big[ \phi_2(X_t; \hat{f}_{t-1}) - \phi_2(X_t; f^*) | \Omega_{t-1} \Big] \Bigg)
$$

$$
\times \Bigg( \phi_1(X_s, A_s, Y_s; \hat{g}_{s-1}, \hat{f}_{s-1}) - \phi_1(X_s, A_s, Y_s; \pi_{s-1}, f^*)
$$

$$
- \mathbb{E} \Big[ \phi_1(X_s, A_s, Y_s; \hat{g}_{s-1}, \hat{f}_{s-1}) - \phi_1(X_s, A_s, Y_s; \pi_{s-1}, f^*) | \Omega_{s-1} \Big]
$$

$$
+ \phi_2(X_s; \hat{f}_{s-1}) - \phi_2(X_s; f^*) - \mathbb{E} \Big[ \phi_2(X_s; \hat{f}_{s-1}) - \phi_2(X_s; f^*) | \Omega_{s-1} \Big] \Bigg) \Bigg].
$$

For $s > t$, we can vanish the covariance terms as

$$
\mathbb{E} \Bigg[ \Bigg( \phi_1(X_t, A_t, Y_t; \hat{g}_{t-1}, \hat{f}_{t-1}) - \phi_1(X_t, A_t, Y_t; \pi_{t-1}, f^*)
$$

$$
- \mathbb{E} \Big[ \phi_1(X_t, A_t, Y_t; \hat{g}_{t-1}, \hat{f}_{t-1}) - \phi_1(X_t, A_t, Y_t; \pi_{t-1}, f^*) | \Omega_{t-1} \Big]
$$

$$
+ \phi_2(X_t; \hat{f}_{t-1}) - \phi_2(X_t; f^*) - \mathbb{E} \Big[ \phi_2(X_t; \hat{f}_{t-1}) - \phi_2(X_t; f^*) | \Omega_{t-1} \Big] \Bigg)
$$

$$
\times \Bigg( \phi_1(X_s, A_s, Y_s; \hat{g}_{s-1}, \hat{f}_{s-1}) - \phi_1(X_s, A_s, Y_s; \pi_{s-1}, f^*)
$$

$$
- \mathbb{E} \Big[ \phi_1(X_s, A_s, Y_s; \hat{g}_{s-1}, \hat{f}_{s-1}) - \phi_1(X_s, A_s, Y_s; \pi_{s-1}, f^*) | \Omega_{s-1} \Big]
$$

$$
+ \phi_2(X_s; \hat{f}_{s-1}) - \phi_2(X_s; f^*) - \mathbb{E} \Big[ \phi_2(X_s; \hat{f}_{s-1}) - \phi_2(X_s; f^*) | \Omega_{s-1} \Big] \Bigg) \Bigg]
$$

$$
= \mathbb{E} \Bigg[ U \mathbb{E} \Bigg[ \Bigg( \phi_1(X_s, A_s, Y_s; \hat{g}_{s-1}, \hat{f}_{s-1}) - \phi_1(X_s, A_s, Y_s; \pi_{s-1}, f^*)
$$

$$
- \mathbb{E} \Big[ \phi_1(X_s, A_s, Y_s; \hat{g}_{s-1}, \hat{f}_{s-1}) - \phi_1(X_s, A_s, Y_s; \pi_{s-1}, f^*) | \Omega_{s-1} \Big]
$$

$$
+ \phi_2(X_s; \hat{f}_{s-1}) - \phi_2(X_s; f^*) - \mathbb{E} \Big[ \phi_2(X_s; \hat{f}_{s-1}) - \phi_2(X_s; f^*) | \Omega_{s-1} \Big] \Bigg) | \Omega_{s-1} \Bigg] \Bigg]
$$

$$
= 0,
$$

where

$$U = \Bigg( \phi_1(X_t, A_t, Y_t; \hat{g}_{t-1}, \hat{f}_{t-1}) - \phi_1(X_t, A_t, Y_t; \pi_{t-1}, f^*)$$

$$- \mathbb{E}\left[ \phi_1(X_t, A_t, Y_t; \hat{g}_{t-1}, \hat{f}_{t-1}) - \phi_1(X_t, A_t, Y_t; \pi_{t-1}, f^*)|\Omega_{t-1} \right]$$

$$+ \phi_2(X_t; \hat{f}_{t-1}) - \phi_2(X_t; f^*)$$

$$- \mathbb{E}\left[ \phi_2(X_t; \hat{f}_{t-1}) - \phi_2(X_t; f^*)|\Omega_{t-1} \right] \Bigg).$$

Therefore, the variance is calculated as

$$\mathrm{Var}\Bigg( \sqrt{T} \frac{1}{T} \sum_{t=1}^{T} \Big\{ \phi_1(X_t, A_t, Y_t; \hat{g}_{t-1}, \hat{f}_{t-1}) - \phi_1(X_t, A_t, Y_t; \pi_{t-1}, f^*)$$

$$- \mathbb{E}\left[ \phi_1(X_t, A_t, Y_t; \hat{g}_{t-1}, \hat{f}_{t-1}) - \phi_1(X_t, A_t, Y_t; \pi_{t-1}, f^*)|\Omega_{t-1} \right]$$

$$+ \phi_2(X_t; \hat{f}_{t-1}) - \phi_2(X_t; f^*)$$

$$- \mathbb{E}\left[ \phi_2(X_t; \hat{f}_{t-1}) - \phi_2(X_t; f^*)|\Omega_{t-1} \right] \Big\} \Bigg)$$

$$= \frac{1}{T} \sum_{t=1}^{T} \mathbb{E}\Bigg[ \bigg( \phi_1(X_t, A_t, Y_t; \hat{g}_{t-1}, \hat{f}_{t-1}) - \phi_1(X_t, A_t, Y_t; \pi_{t-1}, f^*)$$

$$- \mathbb{E}\left[ \phi_1(X_t, A_t, Y_t; \hat{g}_{t-1}, \hat{f}_{t-1}) - \phi_1(X_t, A_t, Y_t; \pi_{t-1}, f^*)|\Omega_{t-1} \right]$$

$$+ \phi_2(X_t; \hat{f}_{t-1}) - \phi_2(X_t; f^*)$$

$$- \mathbb{E}\left[ \phi_2(X_t; \hat{f}_{t-1}) - \phi_2(X_t; f^*)|\Omega_{t-1} \right] \bigg)^2 \Bigg]$$

$$= \frac{1}{T} \sum_{t=1}^{T} \mathbb{E}\Bigg[ \mathbb{E}\bigg[ \Big( \phi_1(X_t, A_t, Y_t; \hat{g}_{t-1}, \hat{f}_{t-1}) - \phi_1(X_t, A_t, Y_t; \pi_{t-1}, f^*)$$

$$- \mathbb{E}\left[ \phi_1(X_t, A_t, Y_t; \hat{g}_{t-1}, \hat{f}_{t-1}) - \phi_1(X_t, A_t, Y_t; \pi_{t-1}, f^*)|\Omega_{t-1} \right]$$

$$+ \phi_2(X_t; \hat{f}_{t-1}) - \phi_2(X_t; f^*)$$

$$- \mathbb{E}\left[ \phi_2(X_t; \hat{f}_{t-1}) - \phi_2(X_t; f^*)|\Omega_{t-1} \right] \Big)^2 |\Omega_{t-1} \bigg] \Bigg]$$

$$= \frac{1}{T} \sum_{t=1}^{T} \mathbb{E}\left[ \mathrm{Var}\bigg( \phi_1(X_t, A_t, Y_t; \hat{g}_{t-1}, \hat{f}_{t-1}) - \phi_1(X_t, A_t, Y_t; \pi_{t-1}, f^*) + \phi_2(X_t; \hat{f}_{t-1}) - \phi_2(X_t; f^*)|\Omega_{t-1} \bigg) \right]$$

$$= \frac{1}{T} \sum_{t=1}^{T} \mathbb{E}\left[ \mathrm{Var}\bigg( \phi_1(X_t, A_t, Y_t; \hat{g}_{t-1}, \hat{f}_{t-1}) - \phi_1(X_t, A_t, Y_t; \pi_{t-1}, f^*)|\Omega_{t-1} \bigg) \right]$$

$$+ \frac{1}{T} \sum_{t=1}^{T} \mathbb{E}\left[ \mathrm{Var}\bigg( \phi_2(X_t; \hat{f}_{t-1}) - \phi_2(X_t; f^*)|\Omega_{t-1} \bigg) \right]$$

$$+ \frac{2}{T} \sum_{t=1}^{T} \mathbb{E}\left[ \mathrm{Cov}\bigg( \phi_1(X_t, A_t, Y_t; \hat{g}_{t-1}, \hat{f}_{t-1}) - \phi_1(X_t, A_t, Y_t; \pi_{t-1}, f^*), \phi_2(X_t; \hat{f}_{t-1}) - \phi_2(X_t; f^*)|\Omega_{t-1} \bigg) \right].$$

Then, we want to show that

$$\frac{1}{T}\sum_{t=1}^{T}\mathbb{E}\left[\operatorname{Var}\left(\phi_1(X_t, A_t, Y_t; \hat{g}_{t-1}, \hat{f}_{t-1}) - \phi_1(X_t, A_t, Y_t; \pi_{t-1}, f^*)|\Omega_{t-1}\right)\right] \to 0, \tag{3}$$

$$\frac{1}{T}\sum_{t=1}^{T}\mathbb{E}\left[\operatorname{Var}\left(\phi_2(X_t; \hat{f}_{t-1}) - \phi_2(X_t; f^*)|\Omega_{t-1}\right)\right] \to 0, \tag{4}$$

$$\frac{2}{T}\sum_{t=1}^{T}\mathbb{E}\left[\operatorname{Cov}\left(\phi_1(X_t, A_t, Y_t; \hat{g}_{t-1}, \hat{f}_{t-1}) - \phi_1(X_t, A_t, Y_t; \pi_{t-1}, f^*), \phi_2(X_t; \hat{f}_{t-1}) - \phi_2(X_t; f^*)|\Omega_{t-1}\right)\right] \to 0 \tag{5}$$

For showing (3)–(5), we consider showing

$$\operatorname{Var}\left(\phi_1(X_t, A_t, Y_t; \hat{g}_{t-1}, \hat{f}_{t-1}) - \phi_1(X_t, A_t, Y_t; \pi_{t-1}, f^*)|\Omega_{t-1}\right) = \mathrm{o}_p(1), \tag{6}$$

$$\operatorname{Var}\left(\phi_2(X_t; \hat{f}_{t-1}) - \phi_2(X_t; f^*)|\Omega_{t-1}\right) = \mathrm{o}_p(1) \tag{7}$$

$$\operatorname{Cov}\left(\phi_1(X_t, A_t, Y_t; \hat{g}_{t-1}, \hat{f}_{t-1}) - \phi_1(X_t, A_t, Y_t; \pi_{t-1}, f^*), \phi_2(X_t; \hat{f}_{t-1}) - \phi_2(X_t; f^*)|\Omega_{t-1}\right) = \mathrm{o}_p(1), \tag{8}$$

The first equation (6) is shown as

$$\operatorname{Var}\left(\phi_1(X_t, A_t, Y_t; \hat{g}_{t-1}, \hat{f}_{t-1}) - \phi_1(X_t, A_t, Y_t; \pi_{t-1}, f^*)|\Omega_{t-1}\right)$$

$$\leq \mathbb{E}\left[\left\{\sum_{a=1}^{K}\frac{\pi^{\mathrm{e}}(a|X_t)\mathbb{1}[A_t = a]\left(Y_t - \hat{f}_{t-1}(a, X_t)\right)}{\hat{g}_{t-1}(a|X_t)} - \sum_{a=1}^{K}\frac{\pi^{\mathrm{e}}(a|X_t)\mathbb{1}[A_t = a]\left(Y_t - f^*(a, X_t)\right)}{\pi_{t-1}(a|X_t)}\right\}^2 |\Omega_{t-1}\right]$$

$$= \mathbb{E}\left[\left\{\sum_{a=1}^{K}\frac{\pi^{\mathrm{e}}(a|X_t)\mathbb{1}[A_t = a]\left(Y_t - \hat{f}_{t-1}(a, X_t)\right)}{\hat{g}_{t-1}(a|X_t)} - \sum_{a=1}^{K}\frac{\pi^{\mathrm{e}}(a|X_t)\mathbb{1}[A_t = a]\left(Y_t - f^*(a, X_t)\right)}{\hat{g}_{t-1}(a|X_t)}\right.\right.$$

$$\left.\left.+ \sum_{a=1}^{K}\frac{\pi^{\mathrm{e}}(a|X_t)\mathbb{1}[A_t = a]\left(Y_t - f^*(a, X_t)\right)}{\hat{g}_{t-1}(a|X_t)} - \sum_{a=1}^{K}\frac{\pi^{\mathrm{e}}(a|X_t)\mathbb{1}[A_t = a]\left(Y_t - f^*(a, X_t)\right)}{\pi_{t-1}(a|X_t)}\right\}^2 |\Omega_{t-1}\right]$$

$$\leq 2\mathbb{E}\left[\left\{\sum_{a=1}^{K}\frac{\pi^{\mathrm{e}}(a|X_t)\mathbb{1}[A_t = a]\left(Y_t - \hat{f}_{t-1}(a, X_t)\right)}{\hat{g}_{t-1}(a|X_t)} - \sum_{a=1}^{K}\frac{\pi^{\mathrm{e}}(a|X_t)\mathbb{1}[A_t = a]\left(Y_t - f^*(a, X_t)\right)}{\hat{g}_{t-1}(a|X_t)}w\right\}^2 z|\Omega_{t-1}\right]$$

$$+ 2\mathbb{E}\left[\left\{\sum_{a=1}^{K}\frac{\pi^{\mathrm{e}}(a|X_t)\mathbb{1}[A_t = a]\left(Y_t - f^*(a, X_t)\right)}{\hat{g}_{t-1}(a|X_t)} - \sum_{a=1}^{K}\frac{\pi^{\mathrm{e}}(a|X_t)\mathbb{1}[A_t = a]\left(Y_t - f^*(a, X_t)\right)}{\pi_{t-1}(a|X_t)}\right\}^2 |\Omega_{t-1}\right]$$

$$\leq 2C\|f^* - \hat{f}_{t-1}\|_2^2 + 2\times 4C^2\|\hat{g}_{t-1} - \pi_{t-1}\|_2^2 = \mathrm{o}_p(1),$$

where $C > 0$ is a constant. Here, we have used a parallelogram law from the third line to the fourth line. We have used $|\hat{f}_{t-1}| < C$, and $0 < \frac{\pi^{\mathrm{e}}}{\pi_{t-1}} < C$, convergence of $\pi_{t-1}$ and convergence rate conditions from the third line to the fourth line. Then, from the $L^r$ convergence theorem (Proposition 2) and the boundedness of the random variables, we can show that as $t \to \infty$,

$$\mathbb{E}\left[\operatorname{Var}\left(\phi_1(X_t, A_t, Y_t; \hat{g}_{t-1}, \hat{f}_{t-1}) - \phi_1(X_t, A_t, Y_t; \pi_{t-1}, f^*)|\Omega_{t-1}\right)\right]$$

$$\leq \mathbb{E}\left[\left|\operatorname{Var}\left(\phi_1(X_t, A_t, Y_t; \hat{g}_{t-1}, \hat{f}_{t-1}) - \phi_1(X_t, A_t, Y_t; \pi_{t-1}, f^*)|\Omega_{t-1}\right)\right|\right] \to 0.$$

Therefore, for any $\epsilon > 0$, there exists a constant $C > 0$ such that

$$\frac{1}{T}\sum_{t=1}^{T}\mathbb{E}\left[\text{Var}\left(\phi_1(X_t, A_t, Y_t; \hat{g}_{t-1}, \hat{f}_{t-1}) - \phi_1(X_t, A_t, Y_t; \pi_{t-1}, f^*)|\Omega_{t-1}\right)\right] \leq C/T + \epsilon.$$

The second equation (7) is derived by Jensen's inequality, and we show (4) as well as (3) by using $L^r$ convergence theorem.

Next, we show the third equation (8) as

$$\text{Cov}\left(\phi_1(X_t, A_t, Y_t; \hat{g}_{t-1}, \hat{f}_{t-1}) - \phi_1(X_t, A_t, Y_t; \pi_{t-1}, f^*), \phi_2(X_t; \hat{f}_{t-1}) - \phi_2(X_t; f^*)|\Omega_{t-1}\right)$$

$$\leq \left|\mathbb{E}\left[\left(\phi_1(X_t, A_t, Y_t; \hat{g}_{t-1}, \hat{f}_{t-1}) - \phi_1(X_t, A_t, Y_t; \pi_{t-1}, f^*)\right)\right.\right.$$
$$\left.\left. - \mathbb{E}\left[\phi_1(X_t, A_t, Y_t; \hat{g}_{t-1}, \hat{f}_{t-1}) - \phi_1(X_t, A_t, Y_t; \pi_{t-1}, f^*)|\Omega_{t-1}\right]\right)\right.$$
$$\left.\times \left(\phi_2(X_t; \hat{f}_{t-1}) - \phi_2(X_t; f^*) - \mathbb{E}\left[\phi_2(X_t; \hat{f}_{t-1}) - \phi_2(X_t; f^*)\right]\right)|\Omega_{t-1}\right]\right|$$

$$\leq \mathbb{E}\left[\left|\left(\phi_1(X_t, A_t, Y_t; \hat{g}_{t-1}, \hat{f}_{t-1}) - \phi_1(X_t, A_t, Y_t; \pi_{t-1}, f^*)\right)\right.\right.$$
$$\left.\left. - \mathbb{E}\left[\phi_1(X_t, A_t, Y_t; \hat{g}_{t-1}, \hat{f}_{t-1}) - \phi_1(X_t, A_t, Y_t; \pi_{t-1}, f^*)|\Omega_{t-1}\right]\right)\right.$$
$$\left.\times \left(\phi_2(X_t; \hat{f}_{t-1}) - \phi_2(X_t; f^*) - \mathbb{E}\left[\phi_2(X_t; \hat{f}_{t-1}) - \phi_2(X_t; f^*)\right]\right)\right||\Omega_{t-1}\right]$$

$$\leq C\mathbb{E}\left[\left|\phi_1(X_t, A_t, Y_t; \hat{g}_{t-1}, \hat{f}_{t-1}) - \phi_1(X_t, A_t, Y_t; \pi_{t-1}, f^*)\right.\right.$$
$$\left.\left. - \mathbb{E}\left[\phi_1(X_t, A_t, Y_t; \hat{g}_{t-1}, \hat{f}_{t-1}) - \phi_1(X_t, A_t, Y_t; \pi_{t-1}, f^*)|\Omega_{t-1}\right]\right||\Omega_{t-1}\right]$$

$$= o_p(1),$$

where $C > 0$ is a constant. From the second to third line, we used Jensen's inequality. From the fourth to fifth line, we used consistencies of $\hat{f}_{t-1}$ and $\hat{g}_{t-1}$, which imply that for all $X_t \in \mathcal{X}$,

$$\phi_1(X_t, A_t, Y_t; \hat{g}_{t-1}, \hat{f}_{t-1}) - \phi_1(X_t, A_t, Y_t; \pi_{t-1}, f^*)$$

$$= \sum_{a=1}^{K}\left(\frac{\pi^e(a|X_t)\mathbb{1}[A_t = a]\left(Y_t - \hat{f}_{t-1}(a, X_t)\right)}{\hat{g}_{t-1}(a|X_t)} - \frac{\pi^e(a|X_t)\mathbb{1}[A_t = a]\left(Y_t - f^*(a, X_t)\right)}{\pi_{t-1}(a|X_t)}\right)$$

$$\leq \sum_{a=1}^{K}\left|\frac{\pi^e(a|X_t)\mathbb{1}[A_t = a]\left(Y_t - \hat{f}_{t-1}(a, X_t)\right)}{\hat{g}_{t-1}(a|X_t)} - \frac{\pi^e(a|X_t)\mathbb{1}[A_t = a]\left(Y_t - f^*(a, X_t)\right)}{\pi_{t-1}(a|X_t)}\right|$$

$$\leq C\sum_{a=1}^{K}\left|\pi_{t-1}(a|X_t)\left(Y_t - \hat{f}_{t-1}(a, X_t)\right) - \hat{g}_{t-1}(a|X_t)\left(Y_t - f^*(a, X_t)\right)\right|$$

$$\leq C\sum_{a=1}^{K}\left|\pi_{t-1}(a|X_t) - \hat{g}_{t-1}(a|X_t)\right|$$

$$- C\sum_{a=1}^{K}\left|\pi_{t-1}(a|X_t)\hat{f}_{t-1}(a, X_t) - \hat{g}_{t-1}(a|X_t)\hat{f}_{t-1}(a, X_t) + \hat{g}_{t-1}(a|X_t)\hat{f}_{t-1}(a, X_t) - \hat{g}_{t-1}(a|X_t)f^*(a, X_t)\right|$$

$$\leq C\sum_{a=1}^{K}\left|\pi_{t-1}(a|X_t) - \hat{g}_{t-1}(a|X_t)\right| - C\sum_{a=1}^{K}\left|\hat{f}_{t-1}(a, X_t) - f^*(a, X_t)\right| = o_p(1),$$

where $C > 0$ is a constant.

Thus, from (3)–(5), the variance of the bias term converges to $0$. Then, from Chebyshev's inequality,

$$
\mathbb{P}\left(\left|\sqrt{T}\frac{1}{T}\sum_{t=1}^{T}\left\{\phi_1(X_t, A_t, Y_t; \hat{g}_{t-1}, \hat{f}_{t-1}) - \phi_1(X_t, A_t, Y_t; \pi_{t-1}, f^*)\right.\right.\right.
$$
$$
- \mathbb{E}\left[\phi_1(X_t, A_t, Y_t; \hat{g}_{t-1}, \hat{f}_{t-1}) - \phi_1(X_t, A_t, Y_t; \pi_{t-1}, f^*)|\Omega_{t-1}\right]
$$
$$
\left.\left.\left.+ \phi_2(X_t; \hat{f}_{t-1}) - \phi_2(X_t; f^*) - \mathbb{E}\left[\phi_2(X_t; \hat{f}_{t-1}) - \phi_2(X_t; f^*)|\Omega_{t-1}\right]\right\}\right| > \varepsilon\right)
$$
$$
\leq \mathrm{Var}\left(\sqrt{T}\frac{1}{T}\sum_{t=1}^{T}\left\{\phi_1(X_t, A_t, Y_t; \hat{g}_{t-1}, \hat{f}_{t-1}) - \phi_1(X_t, A_t, Y_t; \pi_{t-1}, f^*)\right.\right.
$$
$$
- \mathbb{E}\left[\phi_1(X_t, A_t, Y_t; \hat{g}_{t-1}, \hat{f}_{t-1}) - \phi_1(X_t, A_t, Y_t; \pi_{t-1}, f^*)|\Omega_{t-1}\right]
$$
$$
\left.\left.+ \phi_2(X_t; \hat{f}_{t-1}) - \phi_2(X_t; f^*) - \mathbb{E}\left[\phi_2(X_t; \hat{f}_{t-1}) - \phi_2(X_t; f^*)|\Omega_{t-1}\right]\right\}\right)/\varepsilon^2
$$
$$
\to 0.
$$

**Proof of (2).**

$$
\frac{1}{T}\sum_{t=1}^{T}\mathbb{E}\left[\phi_1(X_t, A_t, Y_t; \hat{g}_{t-1}, \hat{f}_{t-1})|\Omega_{t-1}\right] + \frac{1}{T}\sum_{t=1}^{T}\mathbb{E}\left[\phi_2(X_t; \hat{f}_{t-1})|\Omega_{t-1}\right]
$$
$$
- \frac{1}{T}\sum_{t=1}^{T}\mathbb{E}\left[\phi_1(X_t, A_t, Y_t; \pi_{t-1}, f^*)|\Omega_{t-1}\right] - \frac{1}{T}\sum_{t=1}^{T}\mathbb{E}\left[\phi_2(X_t; f^*)|\Omega_{t-1}\right]
$$
$$
= \frac{1}{T}\sum_{t=1}^{T}\mathbb{E}\left[\sum_{a=1}^{K}\frac{\pi^{\mathrm{e}}(a|X_t)\mathbb{1}[A_t = a]\left(Y_t - \hat{f}_{t-1}(a, X_t)\right)}{\hat{g}_{t-1}(a|X_t)}|\Omega_{t-1}\right]
$$
$$
+ \frac{1}{T}\sum_{t=1}^{T}\mathbb{E}\left[\sum_{a=1}^{K}\pi^{\mathrm{e}}(a, X_t)\hat{f}_{t-1}(a, X_t)|\Omega_{t-1}\right]
$$
$$
- \frac{1}{T}\sum_{t=1}^{T}\mathbb{E}\left[\sum_{a=1}^{K}\frac{\pi^{\mathrm{e}}(a|X_t)\mathbb{1}[A_t = a]\left(Y_t - f^*(a, X_t)\right)}{\pi_{t-1}(a|X_t, \Omega_{t-1})}|\Omega_{t-1}\right] \tag{9}
$$
$$
- \frac{1}{T}\sum_{t=1}^{T}\mathbb{E}\left[\sum_{a=1}^{K}\pi^{\mathrm{e}}(a, X_t)f^*(a, X_t)|\Omega_{t-1}\right].
$$

Because (9) is 0,

$$\frac{1}{T}\sum_{t=1}^{T}\mathbb{E}\left[\phi_1(X_t, A_t, Y_t; \hat{g}_{t-1}, \hat{f}_{t-1})|\Omega_{t-1}\right] + \frac{1}{T}\sum_{t=1}^{T}\mathbb{E}\left[\phi_2(X_t; \hat{f}_{t-1})|\Omega_{t-1}\right]$$

$$- \frac{1}{T}\sum_{t=1}^{T}\mathbb{E}\left[\phi_1(X_t, A_t, Y_t; \pi_{t-1}, f^*)|\Omega_{t-1}\right] - \frac{1}{T}\sum_{t=1}^{T}\mathbb{E}\left[\phi_2(X_t; f^*)|\Omega_{t-1}\right]$$

$$= \frac{1}{T}\sum_{t=1}^{T}\mathbb{E}\left[\sum_{a=1}^{K}\frac{\pi^{\mathrm{e}}(a|X_t)\mathbb{1}[A_t=a]\left(Y_t - \hat{f}_{t-1}(a, X_t)\right)}{\hat{g}_{t-1}(a|X_t)}|\Omega_{t-1}\right]$$

$$+ \frac{1}{T}\sum_{t=1}^{T}\mathbb{E}\left[\sum_{a=1}^{K}\pi^{\mathrm{e}}(a, X_t)\hat{f}_{t-1}(a, X_t)|\Omega_{t-1}\right] - \frac{1}{T}\sum_{t=1}^{T}\mathbb{E}\left[\sum_{a=1}^{K}\pi^{\mathrm{e}}(a, X_t)f^*(a, X_t)|\Omega_{t-1}\right]$$

$$= \frac{1}{T}\sum_{t=1}^{T}\mathbb{E}\left[\sum_{a=1}^{K}\frac{\pi^{\mathrm{e}}(a|X_t)\mathbb{1}[A_t=a]\left(Y_t - \hat{f}_{t-1}(a, X_t)\right)}{\hat{g}_{t-1}(a|X_t)}|\Omega_{t-1}\right]$$

$$- \frac{1}{T}\sum_{t=1}^{T}\mathbb{E}\left[\sum_{a=1}^{K}\pi^{\mathrm{e}}(a, X_t)\left(f^*(a, X_t) - \hat{f}_{t-1}(a, X_t)\right)|\Omega_{t-1}\right]$$

$$= \frac{1}{T}\sum_{t=1}^{T}\sum_{a=1}^{K}\mathbb{E}\left[\mathbb{E}\left[\frac{\pi^{\mathrm{e}}(a|X_t)\pi_{t-1}(a|X_t, \Omega_{t-1})\left(f^*(a, X_t) - \hat{f}_{t-1}(a, X_t)\right)}{\hat{g}_{t-1}(a|X_t)}\right.\right.$$

$$\left.\left. - \pi^{\mathrm{e}}(a, X_t)\left(f^*(a, X_t) - \hat{f}_{t-1}(a, X_t)\right)|X_t, \Omega_{t-1}\right]|\Omega_{t-1}\right]$$

$$\leq \frac{1}{T}\sum_{t=1}^{T}\sum_{a=1}^{K}\left|\mathbb{E}\left[\frac{\pi^{\mathrm{e}}(a|X_t)\left(\pi_{t-1}(a|X_t) - \hat{g}_{t-1}(a|X_t)\right)\left(f^*(a, X_t) - \hat{f}_{t-1}(a, X_t)\right)}{\hat{g}_{t-1}(a|X_t)}|\Omega_{t-1}\right]\right|.$$

By using Hölder's inequality $\|fg\|_1 \leq \|f\|_2\|g\|_2$, for a constant $C > 0$, we have

$$\leq \frac{C}{T}\sum_{t=1}^{T}\left\|\pi_{t-1}(a|X_t, \Omega_{t-1}) - \hat{g}_{t-1}(a|X_t)\right\|_2\left\|f^*(a, X_t) - \hat{f}_{t-1}(a, X_t)\right\|_2$$

$$= \frac{C}{T}\sum_{t=1}^{T}\mathrm{o}_p(t^{-p})\mathrm{o}_p(t^{-q})$$

$$= \frac{C}{T}\sum_{t=1}^{T}\mathrm{o}_p(t^{-1/2}).$$

$\square$

# C  Adaptive-fitting and batched samples

Section C.1 supplements the description of adaptive-fitting. Next, in Section C.2, we introduce the AIPW estimator when the samples are given in batch form, and the true logging policy is given $\pi_t$. This is essentially the same as the generalized method of moment (GMM), which gives an asymptotically normal estimator for martigaale difference sequences (MDS), and we do not use adaptive-fitting. Based on this estimator, in Section C.3, we introduce the ADR estimator when the data is given in batch form, but the true logging policy $\pi_t$ is not given.

## C.1  Details of adaptive-fitting

As Section 5, let us define the parameter of interest $\theta_0$ that satisfies $\mathbb{E}[\psi(W_t; \theta_0, \eta_0)] = 0$, where $\{W_t\}_{t=1}^{T}$ are observations, $\eta_0$ is a nuisance parameter, and $\psi$ is a score function. Let us define two

estimators $\hat{\theta}_T$ and $\check{\theta}_T$ as $\frac{1}{T}\sum_{t=1}^{T}\psi(W_t, \hat{\theta}_T, \eta_{t-1}) = 0$ and $\frac{1}{T}\sum_{t=1}^{T}\psi(W_t, \check{\theta}_T, \eta_0) = 0$. Suppose that $\check{\theta}_T$ is an asymptotically normal estimator of $\theta_0$. Then, if $\check{\theta}_T - \hat{\theta}_T$ converges to 0 with convergence rate $o_p(1/\sqrt{T})$, $\hat{\theta}_T$ is also an asymptotically normal estimator. In general, we cannot obtain such a fast convergence rate. However, by using double robustness, we can obtain the convergence rate. In ADR estimator, this conditions appears as $\|\hat{g}_{t-1}(a|X_t) - \pi_{t-1}(a|X_t, \Omega_{t-1})\|_2 = o_p(t^{-p})$, and $\|\hat{f}_{t-1}(a, X_t) - f^*(a, X_t)\|_2 = o_p(t^{-q})$, where $p, q > 0$ such that $p + q = 1/2$, and the expectation of the norm is taken over $X_t$. This allows us to obtain $o_p(1/\sqrt{T})$ of the asympttoic bias. The image of the vanishing asymptotic bias is shown in Figure 3.

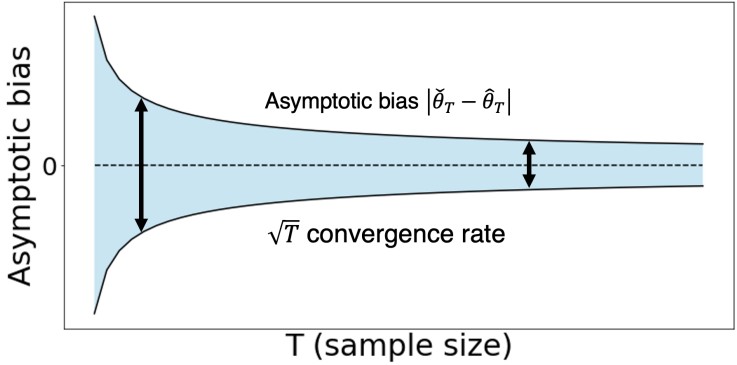

Figure 3: Convergence of asymptotic bias $|\check{\theta}_T - \hat{\theta}_T|$.

## C.2 AIPW estimator with batched samples when the true logging policy is known

The proposed adaptive-fitting can be applied when batched samples are given. Let $M$ denote the number of batch updates and $\tau \in I = \{1, 2, \ldots, M\}$ denotes the batch index. For $\tau \in I$, the probability is updated at a period $t_\tau$, where $t_\tau - t_{\tau-1} = Tr_\tau$, using samples $\{(X_t, Y_t, A_t)\}_{t=t_{\tau-1}}^{t_\tau}$, where $r_1 + r_2 + \cdots + r_M = 1$ and $t_0 = 0$. Thus, in addition to the DGP, we assume that

$$\{(X_t, A_t, Y_t)\}_{t=t_{\tau-1}}^{t_\tau} \overset{i.i.d.}{\sim} p(x)\pi_\tau(a|x, \Omega_{t_{\tau-1}})p_a(y|x),$$

where $\pi_\tau(a|x, \Omega_{t_{\tau-1}})$ denotes the probability of choosing an action updated based on samples until the period $t_{\tau-1}$.

We consider asymptotic properties based on the assumption of $t_\tau - t_{\tau-1} \to \infty$ as $T \to \infty$ for fixed $\tau$. This strategy is the same as Zhang et al. (2020). Because $\{(X_t, A_t, Y_t)\}_{t=t_{\tau-1}}^{t_\tau}$ is i.i.d., we can use the standard limit theorems for the partial sum of the samples to obtain an asymptotically normal estimator of $\theta_0 = R(\pi^e)$. However, we also have the motivation to use all samples together to increase the efficiency of the estimator. Therefore, by using the GMM, we propose an estimator of $\theta_0$ considering the sample averages of each block as an empirical moment conditions. Although we cannot use standard CLT, we can can apply the CLT for the MDS by appropriately constructing an estimator.

For an index of batch $\tau \in I$, a function $f \in \mathcal{F}$ such that $f : \mathcal{A} \times \mathcal{X} \to \mathbb{R}$ and an evaluation policy $\pi^e \in \Pi$, we define a function $h_t$ as

$$h_t(x, k, y; \tau, R, f, \pi_\tau, \pi^e) = \frac{1}{r_\tau}\xi_t(x, k, y; \tau, R, f, \pi_\tau, \pi^e)\mathbb{1}\left[t_{\tau-1} < t \le t_\tau\right],$$

where $R \in \mathbb{R}$, $\xi_t(x, k, y; \tau, R, f, \pi_\tau, \pi^e) := \phi_t(x, k, y; \tau, f, \pi_\tau, \pi^e) - R$ and

$$\phi_t(x, k, y; \tau, f, \pi_\tau, \pi^e) := \sum_{a=1}^{K}\pi^e(a|x)\left\{\frac{\mathbb{1}[k=a]\{y - f(a, x)\}}{\pi_\tau(a|x, \Omega_{t_{\tau-1}})} + f(a, x)\right\}.$$

Let us note that the sequence $\left\{h_t(X_t, A_t, Y_t; \tau, R(\pi^{\mathrm{e}}), \hat{f}_{t-1}, \pi_\tau, \pi^{\mathrm{e}})\right\}_{t=1}^T$ is an MDS: for $h_t(X_t, A_t, Y_t; \tau, R(\pi^{\mathrm{e}}), \hat{f}_{t-1}, \pi_\tau, \pi^{\mathrm{e}})$, by using $\mathbb{E}[\mathbb{1}[A_t = a]|\mathbb{H}_{t-1}] = \pi_\tau(a|X_t, \Omega_{t_{\tau-1}})$, we have

$$
\mathbb{E}\left[h_t(X_t, A_t, Y_t; \tau, R(\pi^{\mathrm{e}}), \hat{f}_{t-1}, \pi_\tau, \pi^{\mathrm{e}})|\Omega_{t-1}\right]
$$
$$
= \mathbb{E}\left[\frac{\mathbb{1}\left[t_{\tau-1} < t \le t_\tau\right]}{r_\tau}\xi_t(x, k, y; \tau, R(\pi^{\mathrm{e}}), \hat{f}_{t-1}, \pi_\tau, \pi^{\mathrm{e}})|\Omega_{t-1}\right]
$$
$$
= \frac{\mathbb{1}\left[t_{\tau-1} < t \le t_\tau\right]}{r_\tau}\mathbb{E}\left[\xi_t(x, k, y; \tau, R(\pi^{\mathrm{e}}), \hat{f}_{t-1}, \pi_\tau, \pi^{\mathrm{e}})|\Omega_{t-1}\right]
$$
$$
= \frac{\mathbb{1}\left[t_{\tau-1} < t \le t_\tau\right]}{r_\tau} \times 0 = 0.
$$

Let us also define

$$
\boldsymbol{h}_t\left(X_t, A_t, Y_t; R, \hat{f}_{t-1}, \pi_\tau, \pi^{\mathrm{e}}\right) := \begin{pmatrix} h_t(X_t, A_t, Y_t; 1, R, \hat{f}_{t-1}, \pi_1, \pi^{\mathrm{e}}) \\ h_t(X_t, A_t, Y_t; 2, R, \hat{f}_{t-1}, \pi_2, \pi^{\mathrm{e}}) \\ \vdots \\ h_t(X_t, A_t, Y_t; M, R, \hat{f}_{t-1}, \pi_M, \pi^{\mathrm{e}}) \end{pmatrix}.
$$

Then, the sequence $\left\{\boldsymbol{h}_t\left(X_t, A_t, Y_t; R(\pi^{\mathrm{e}}), \hat{f}_{t-1}, \pi_\tau, \pi^{\mathrm{e}}\right)\right\}_{t=1}^T$ is an MDS with respect to $\{\Omega_t\}_{t=0}^{T-1}$, i.e.,

$$
\mathbb{E}\left[\boldsymbol{h}_t\left(X_t, A_t, Y_t; R(\pi^{\mathrm{e}}), \hat{f}_{t-1}, \pi_\tau, \pi^{\mathrm{e}}\right)|\Omega_{t-1}\right] = \boldsymbol{0}.
$$

Using the sequence $\left\{\boldsymbol{h}_t\left(X_t, A_t, Y_t; R, \hat{f}_{t-1}, \pi_\tau, \pi^{\mathrm{e}}\right)\right\}_{t=1}^T$, we define an estimator of $R(\pi^{\mathrm{e}})$ as

$$
\hat{R}_T^{\mathrm{batch}}(\pi^{\mathrm{e}}) = \arg\min_{R \in \mathbb{R}} \left(\hat{\boldsymbol{q}}_T(R)\right)^\top \hat{W}_T \left(\hat{\boldsymbol{q}}_T(R)\right), \tag{10}
$$

where $\hat{\boldsymbol{q}}_T(R) = \frac{1}{T}\sum_{t=1}^T \boldsymbol{h}_t\left(X_t, A_t, Y_t; R, \hat{f}_{t-1}, \pi_\tau, \pi^{\mathrm{e}}\right)$ and $\hat{W}_T$ is a data-dependent $(M \times M)$-dimensional positive semi-definite matrix. Let us note that the estimator defined in Eq. (10) is an application of GMM with the moment condition

$$
\boldsymbol{q}(R(\pi^{\mathrm{e}})) = \mathbb{E}\left[\frac{1}{T}\sum_{t=1}^T \boldsymbol{h}_t\left(X_t, A_t, Y_t; R(\pi^{\mathrm{e}}), \hat{f}_{t-1}, \pi_\tau, \pi^{\mathrm{e}}\right)\right] = 0.
$$

For the minimization problem defined in Eq. (10), we can analytically calculate the minimizer as

$$
\hat{R}_T^{\mathrm{batch}}(\pi^{\mathrm{e}}) = w_T^\top D_T(\pi^{\mathrm{e}}),
$$

where $w_T = (w_{T,1} \ \cdots \ w_{T,M})^\top$ is an $M$-dimensional vector such that $\sum_{\tau=1}^M w_{T,\tau} = 1$, and

$$
D_T(\pi^{\mathrm{e}}) = \begin{pmatrix} \frac{1}{t_1}\sum_{t=1}^{t_1} \phi_t(X_t, A_t, Y_t; 1, f_{t-1}, \pi_1, \pi^{\mathrm{e}}) \\ \frac{1}{t_2-t_1}\sum_{t=t_1+1}^{t_2} \phi_t(X_t, A_t, Y_t; 2, f_{t-1}, \pi_2, \pi^{\mathrm{e}}) \\ \vdots \\ \frac{1}{T-t_{M-1}}\sum_{t=t_{M-1}+1}^{T} \phi_t(X_t, A_t, Y_t; M, f_{t-1}, \pi_M, \pi^{\mathrm{e}}) \end{pmatrix}.
$$

Here, we show the asymptotic normality of the proposed estimator $\hat{\theta}_T^{\mathrm{OPE}}$.

**Theorem 3** (Asymptotic distribution the proposed estimator). *Suppose that*

**(i)** $w_T = (w_{T,1} \ \cdots \ w_{T,M})^\top \xrightarrow{\mathrm{P}} w = (w_1 \ \cdots \ w_M)^\top$;

**(ii)** $w_{T,\tau} > 0$ *and* $\sum_{\tau=1}^M w_{T,\tau} = 1$.

*Then, under Assumptions 1, 3, 5, 7,*

$$\sqrt{T}\big(\hat{R}_T^{\text{batch}}(\pi^{\text{e}}) - R(\pi^{\text{e}})\big) \xrightarrow{\text{d}} \mathcal{N}\big(0, \sigma^2\big),$$

*where $\sigma^2 = \sum_{\tau=1}^{M} w_\tau \Psi(\pi^{\text{e}}, \pi_\tau)$.*

In the proposed method, we construct a moment condition using martingale difference sequences. On the other hand, for some readers, using martingale difference sequences may look unnecessary because samples are i.i.d in each block between $t_{\tau-1}$ and $t_\tau$. Therefore, such readers also might feel that we can use $f_T(a, x)$, which is an estimator of $\mathbb{E}[Y(a) \mid x]$ using samples until $T$-th period, without going through constructing several estimators $\{f_{t_\tau}\}_{\tau=0}^{M-1}$. However, in that case, it is difficult to guarantee the asymptotic normality of the proposed estimator. For example, we can consider Cramér-Wold theorem to consider this problem. This motivations shares with the GMM and the method of Zhang et al. (2020).

The proof of Theorem 3 is shown as follows.

**Choice of weight $w_T$.** We discuss the choice of weight $w_T$. A naive choice is weighting the moment conditions equality; that is, $w_{\tau,T} = \frac{1}{M}$. Next, we consider an efficient weight $w_T$ that minimizes the asymptotic variance of $\hat{R}_T^{\text{batch}}(\pi^{\text{e}})$. In the GMM, the $\tau$-th element of the efficient weight is given as $w_\tau^* = \frac{1}{\Psi(\pi^{\text{e}},\pi_\tau)} / \sum_{\tau'=1}^{M} \frac{1}{\Psi(\pi^{\text{e}},\pi_{\tau'})}$ (Hamilton, 1994). Here, we use the orthogonality among moment conditions; that is, zero covariance. In this case, the asymptotic variance becomes $1 / \sum_{\tau'=1}^{M} \frac{1}{\sigma_{\tau'}^2}$. Therefore, for gaining efficiency, we use a weight $\hat{w}_{T,\tau} = \frac{1}{\hat{\Psi}_T(\pi^{\text{e}},\pi_\tau)} / \sum_{\tau'=1}^{M} \frac{1}{\hat{\Psi}_T(\pi^{\text{e}},\pi_{\tau'})}$, where $\hat{\Psi}_T(\pi^{\text{e}}, \pi_\tau)$ is an estimator of $\Psi(\pi^{\text{e}}, \pi_\tau)$.

**Proof of Theorem 3.** Instead of $\hat{R}_T^{\text{batch}}(\pi^{\text{e}}) = w_T D_T(\pi^{\text{e}})$, from the original formulation Eq. (10), we consider an estimator $\hat{R}_T^{\text{batch}}(\pi^{\text{e}}) = \big(I^\top \hat{W}_T I\big)^{-1} I^\top \hat{W}_T D_T(\pi^{\text{e}})$, where $W_T$ is a $(M \times M)$-dimensional positive-definite matrix. Let us note that $w_T = \big(I^\top \hat{W}_T I\big)^{-1} I^\top \hat{W}_T$. We prove the following theorem, which is a generalized statement of Theorem 3.

**Theorem 4** (Asymptotic distribution the AIPW estimator under batch update)**.** *Suppose that*

**(i)** $\hat{W}_T \xrightarrow{\text{p}} W$;

**(ii)** $W$ *is a positive definite;*

*Then, under Assumptions 1, 3, 5, 7,*

$$\sqrt{T}\big(\hat{R}_T^{\text{batch}}(\pi^{\text{e}}) - R(\pi^{\text{e}})\big) \xrightarrow{\text{d}} \mathcal{N}\big(0, \sigma^2\big),$$

*where $\sigma^2 = \big(I^\top W I\big)^{-1} I^\top W \Sigma W^\top I \big(I^\top W I\big)^{-1}$ and $\Sigma$ is a $(M \times M)$ diagonal matrix such that the $(\tau \times \tau)$-element is $\Psi(\pi^{\text{e}}, \pi_\tau)$.*

*Proof.* Let us define $\hat{\boldsymbol{q}}_T(R(\pi^{\text{e}})) = (\hat{q}_{1,T}(R(\pi^{\text{e}})) \ \hat{q}_{2,T}(R(\pi^{\text{e}})) \ \cdots \ \hat{q}_{M,T}(R(\pi^{\text{e}})))$, where
$\hat{q}_{\tau,T}(R(\pi^{\text{e}})) =$

$$\frac{1}{T}\frac{1}{r_1} \sum_{t=1}^{T} \left( \sum_{a=1}^{K} \left\{ \frac{\pi^{\text{e}}(a \mid X_t)\mathbb{1}[A_t = a]\{Y_t - \hat{f}_{t-1}(a, X_t)\}}{\pi_1(a \mid x, \Omega_0)} + \pi^{\text{e}}(a \mid X_t)\hat{f}_{t-1}(a, X_t) \right\} - R(\pi^{\text{e}}) \right)$$
$$\times \mathbb{1}\big[t_0 = 0 < t \le t_1\big].$$

For $\sqrt{T}\big(\hat{R}_T^{\text{batch}}(\pi^{\text{e}}) - R(\pi^{\text{e}})\big) = \big(I^\top \hat{W}_T I\big)^{-1} I^\top \hat{W}_T \sqrt{T}\hat{\boldsymbol{q}}_T(R(\pi^{\text{e}}))$, we show that

$$\sqrt{T}\hat{\boldsymbol{q}}_T(R(\pi^{\text{e}})) \xrightarrow{\text{d}} \mathcal{N}\big(0, \Sigma\big),$$

where $\Sigma$ is a diagonal matrix such that the $(\tau, \tau)$-element is

$$\frac{1}{r_\tau}\mathbb{E}\left[ \sum_{a=1}^{K} \frac{\big(\pi^{\text{e}}(a \mid X)\big)^2 \text{Var}(Y(a) \mid X)}{\pi_\tau(a \mid X, \Omega_{t_{\tau-1}})} + \left( \sum_{a=1}^{K} \pi^{\text{e}}(a \mid X)\mathbb{E}\big[Y(a) \mid X\big] - R(\pi^{\text{e}}) \right)^2 \right].$$

Then, from Slutsky Theorem, we can show that

$$\left(I^\top \hat{W}_T I\right)^{-1} I^\top \hat{W}_T \sqrt{T} \hat{q}_T(R(\pi^e)) \xrightarrow{d} \mathcal{N}\left(0, \left(I^\top W I\right)^{-1} I^\top W \Sigma W^\top I (I^\top W I)^{-1}\right).$$

To show this result, we use the CLT for MDS by checking the following conditions:

**(a)** $(1/T)\sum_{t=1}^T \Sigma_t \to \Sigma$, where

$$\Sigma_t = \mathbb{E}\left[\left(h_t\left(X_t, A_t, Y_t; R(\pi^e), f_{t-1}, \pi_\tau, \pi^e\right)\right)\left(h_t\left(X_t, A_t, Y_t; R(\pi^e), f_{t-1}, \pi_\tau, \pi^e\right)\right)^\top\right];$$

**(b)** $\mathbb{E}\big[\tilde{h}_t(i, R(\pi^e), f_{t-1}, \pi^e)\tilde{h}_t(j, R(\pi^e), f_{t-1}, \pi^e)\tilde{h}_t(a, R(\pi^e), f_{t-1}, \pi^e)\tilde{h}_t(l, R(\pi^e), f_{t-1}, \pi^e)\big] <$ $\infty$ for $i, j, k, l \in I$, where $\tilde{h}_t(a, R(\pi^e), f_{t-1}, \pi^e) = h_t^{\mathrm{OPE}}(X_t, A_t, Y_t; k, R(\pi^e), f_k, \pi^e)$ for $k \in I$;

**(c)** $\frac{1}{T}\sum_{t=1}^T \left(h_t\left(X_t, A_t, Y_t; R(\pi^e), f_{t-1}, \pi_\tau, \pi^e\right)\right)\left(h_t\left(X_t, A_t, Y_t; R(\pi^e), f_{t-1}, \pi_\tau, \pi^e\right)\right)^\top \xrightarrow{p} \Sigma,$

Note that the GMM shows the asymptotic normality for more general cases.

**Step 1: Condition (a)**

From

$$\Sigma_t = \mathbb{E}\left[\left(h_t\left(X_t, A_t, Y_t; R(\pi^e), f_{t-1}, \pi_\tau, \pi^e\right)\right)\left(h_t\left(X_t, A_t, Y_t; R(\pi^e), f_{t-1}, \pi_\tau, \pi^e\right)\right)^\top\right],$$

the matrix $(1/T)\sum_{t=1}^T \Omega_t$ becomes a diagonal matrix such that the $(\tau, \tau)$-element is

$$\frac{1}{r_\tau^2 T}\sum_{t=1}^T \mathbb{E}\left[\left(\sum_{a=1}^K \left\{\frac{\pi^e(a \mid X_t)\mathbb{1}[A_t = a]\{Y_t - f_{t-1}(a, X_t)\}}{\pi_\tau(a \mid X_t, \Omega_{t_{\tau-1}})} + \pi^e(a \mid X_t)f_{t-1}(a, X_t)\right\} - R(\pi^e)\right)^2 \right.$$
$$\left. \times \mathbb{1}\big[t_{\tau-1} < t \le t_\tau\big]\right].$$

For $\tau \in I$ and $t$ such that $t_{\tau-1} < t \le t_\tau$,

$$\mathbb{E}\left[\left(\sum_{a=1}^K \left\{\frac{\pi^e(a \mid X_t)\mathbb{1}[A_t = a]\{Y_t - f_{t-1}(a, X_t)\}}{\pi_\tau(a \mid X_t, \Omega_{t_{\tau-1}})} + \pi^e(a \mid X_t)f_{t-1}(a, X_t)\right\} - R(\pi^e)\right)^2\right]$$

$$- \mathbb{E}\left[\left(\sum_{a=1}^K \left\{\frac{\pi^e(a \mid X_t)\mathbb{1}[A_t = a]\{Y_t - \mathbb{E}[Y_t(a) \mid X_t]\}}{\pi_\tau(a \mid X_t, \Omega_{t_{\tau-1}})} + \pi^e(a \mid X_t)\mathbb{E}[Y(a) \mid X_t]\right\} - R(\pi^e)\right)^2\right]$$

$$\le \mathbb{E}\left[\left|\left(\sum_{a=1}^K \left\{\frac{\pi^e(a \mid X_t)\mathbb{1}[A_t = a]\{Y_t - f_{t-1}(a, X_t)\}}{\pi_\tau(a \mid X_t, \Omega_{t_{\tau-1}})} + \pi^e(a \mid X_t)f_{t-1}(a, X_t)\right\} - R(\pi^e)\right)^2\right.\right.$$

$$\left.\left.- \left(\sum_{a=1}^K \left\{\frac{\pi^e(a \mid X_t)\mathbb{1}[A_t = a]\{Y_t - \mathbb{E}[Y_t(a) \mid X_t]\}}{\pi_\tau(a \mid X_t, \Omega_{t_{\tau-1}})} + \pi^e(a \mid X_t)\mathbb{E}[Y(a) \mid X_t]\right\} - R(\pi^e)\right)^2\right|\right]$$

Because $\alpha^2 - \beta^2 = (\alpha + \beta)(\alpha - \beta)$, there exists a constant $\gamma_0 > 0$ such that

$$\le \gamma_0 \mathbb{E}\left[\left|\sum_{a=1}^K \left\{\frac{\pi^e(a \mid X_t)\mathbb{1}[A_t = a]\{Y_t - f_{t-1}(a, X_t)\}}{\pi_\tau(a \mid X_t, \Omega_{t_{\tau-1}})} + \pi^e(a \mid X_t)f_{t-1}(a, X_t)\right.\right.\right.$$

$$\left.\left.\left.- \frac{\pi^e(a \mid X_t)\mathbb{1}[A_t = a]\{Y_t - \mathbb{E}[Y_t(a) \mid X_t]\}}{\pi_\tau(a \mid X_t, \Omega_{t_{\tau-1}})} - \pi^e(a \mid X_t)\mathbb{E}[Y(a) \mid X_t]\right\}\right|\right]$$

Then, there exist constants $\gamma_1 > 0$ such that

$$\leq \gamma_1 \mathbb{E}\left[\sum_{a=1}^K \left| f_{t-1}(a, X_t) - \mathbb{E}\big[Y(a) \mid X_t\big]\right|\right].$$

Here, from the assumption that $f_{t-1}(a, x) - \mathbb{E}\big[Y(a) \mid X\big] \xrightarrow{\mathrm{P}} 0$ for $\tau = 2, 3, \ldots, M$, and $f_{t_{\tau-1}}(a, x)$ is bounded for $\tau \in I$, we can use $L^r$ convergence theorem. First, to use $L^r$ convergence theorem, we use boundedness of $f_{t_m}$ to derive the uniform integrability of $f_{t_m}$ for $m = 0, 1, \ldots, \tau - 1$. Then, from $L^r$ convergence theorem, we have $\mathbb{E}\big[|f_{t_m}(a, X) - \mathbb{E}[Y(a) \mid X]|\big] \to 0$ as $t_m \to \infty$. Using this results, we can show that, as $t_{\tau-1} \to \infty$ (this also means $T \to \infty$),

$$\gamma_1 \sum_{a=1}^K \mathbb{E}\left[\left| f_{t-1}(a, X_t) - \mathbb{E}\big[Y(a) \mid X_t\big]\right|\right] \to 0.$$

Therefore, as $t_{\tau-1} \to \infty$ ($T \to \infty$),

$$\mathbb{E}\left[\left(\sum_{a=1}^K \left\{\frac{\pi^{\mathrm{e}}(a \mid X_t)\mathbb{1}[A_t = a]\{Y_t - f_{t-1}(a, X_t)\}}{\pi_\tau(a \mid X_t, \Omega_{t_{\tau-1}})} + \pi^{\mathrm{e}}(a \mid X_t)f_{t-1}(a, X_t)\right\} - R(\pi^{\mathrm{e}})\right)^2\right]$$

$$\to \mathbb{E}\left[\left(\sum_{a=1}^K \left\{\frac{\pi^{\mathrm{e}}(a \mid X_t)\mathbb{1}[A_t = a]\{Y_t - \mathbb{E}[Y_t(a) \mid X_t]\}}{\pi_\tau(a \mid X_t, \Omega_{t_{\tau-1}})} + \pi^{\mathrm{e}}(a \mid X_t)\mathbb{E}[Y(a) \mid X_t]\right\} - R(\pi^{\mathrm{e}})\right)^2\right].$$

Then, by using $\mathbb{1}[A_t = a]\mathbb{1}[A_t = l] = 0$, $\mathbb{E}\left[\frac{\mathbb{1}[A_t=a]Y_t^2}{\left(\pi_\tau(a|X_t,\Omega_{t_{\tau-1}})\right)^2}\right] = \mathbb{E}\left[\frac{\mathbb{E}\left[Y_t^2(a)|X_t\right]}{\pi_\tau(a|X_t,\Omega_{t_{\tau-1}})}\right]$, and

$\frac{1}{r_\tau T}\sum_{t=1}^T \mathbb{1}\big[t_{\tau-1} < t \leq t_\tau\big] = 1$,

$$\mathbb{E}\left[\left(\sum_{a=1}^K \left\{\frac{\pi^{\mathrm{e}}(a \mid X_t)\mathbb{1}[A_t = a]\{Y_t - \mathbb{E}[Y_t(a) \mid X_t]\}}{\pi_\tau(a \mid X_t, \Omega_{t_{\tau-1}})} + \pi^{\mathrm{e}}(a \mid X_t)\mathbb{E}[Y(a) \mid X_t]\right\} - R(\pi^{\mathrm{e}})\right)^2\right]$$

$$= \mathbb{E}\left[\sum_{a=1}^K \left\{\frac{\left(\pi^{\mathrm{e}}(a \mid X_t)\right)^2\mathrm{Var}\big(Y_t(a) \mid X_t\big)}{\pi_\tau(a \mid X_t, \Omega_{t_{\tau-1}})} + \left(\pi^{\mathrm{e}}(a \mid X_t)\mathbb{E}\big[Y_t(a) \mid X_t\big] - R(\pi^{\mathrm{e}})\right)^2\right\}\right].$$

In addition, the variance does not depend on $t$. We represent the independence by omitting the subscript $t$, i.e.,

$$\mathbb{E}\left[\sum_{a=1}^K \left\{\frac{\left(\pi^{\mathrm{e}}(a \mid X_t)\right)^2\mathrm{Var}\big(Y_t(a) \mid X_t\big)}{\pi_\tau(a \mid X_t, \Omega_{t_{\tau-1}})} + \left(\pi^{\mathrm{e}}(a \mid X_t)\mathbb{E}\big[Y_t(a) \mid X_t\big] - R(\pi^{\mathrm{e}})\right)^2\right\}\right]$$

$$= \mathbb{E}\left[\sum_{a=1}^K \left\{\frac{\left(\pi^{\mathrm{e}}(a \mid X)\right)^2\mathrm{Var}\big(Y(a) \mid X\big)}{\pi_\tau(a \mid X, \Omega_{t_{\tau-1}})} + \left(\pi^{\mathrm{e}}(a \mid X)\mathbb{E}\big[Y_t(a) \mid X\big] - R(\pi^{\mathrm{e}})\right)^2\right\}\right].$$

Therefore, we have

$$\frac{1}{r_\tau^2 T}\sum_{t=1}^T \mathbb{E}\left[\left(\sum_{a=1}^K \left\{\frac{\pi^{\mathrm{e}}(a \mid X_t)\mathbb{1}[A_t = a]\{Y_t - f_{t-1}(a, X_t)\}}{\pi_\tau(a \mid X_t, \Omega_{t_{\tau-1}})} + \pi^{\mathrm{e}}(a \mid X_t)f_{t-1}(a, X_t)\right\} - R(\pi^{\mathrm{e}})\right)^2\right.$$

$$\left.\times \mathbb{1}\big[t_{\tau-1} < t \leq t_\tau\big]\right]$$

$$\to \frac{1}{r_\tau}\mathbb{E}\left[\sum_{a=1}^K \left\{\frac{\left(\pi^{\mathrm{e}}(a \mid X)\right)^2\mathrm{Var}\big(Y(a) \mid X\big)}{\pi_\tau(a \mid X, \Omega_{t_{\tau-1}})} + \left(\pi^{\mathrm{e}}(a \mid X)\mathbb{E}\big[Y(a) \mid X\big] - R(\pi^{\mathrm{e}})\right)^2\right\}\right].$$

Thus, the matrix $(1/T)\sum_{t=1}^T \Sigma_t$ converges to a diagonal matrix $\Sigma$ as $T \to \infty$, where the $(\tau, \tau)$-element of $\Sigma$ is

$$\frac{1}{r_\tau}\mathbb{E}\left[\sum_{a=1}^K \left\{\frac{\left(\pi^{\mathrm{e}}(a \mid X)\right)^2\mathrm{Var}\big(Y(a) \mid X\big)}{\pi_\tau(a \mid X, \Omega_{t_{\tau-1}})} + \left(\pi^{\mathrm{e}}(a \mid X)\mathbb{E}\big[Y(a) \mid X\big] - R(\pi^{\mathrm{e}})\right)^2\right\}\right].$$

**Step 2: Condition (b)**

Because we assume that all variables are bounded, this condition holds.

**Step 3: Condition (c)**

Here, we check that $(1/T)\sum_{t=1}^{T}\left(\boldsymbol{h}_t\left(X_t,A_t,Y_t;R(\pi^{\mathrm{e}}),f_{t-1},\pi_\tau,\pi^{\mathrm{e}}\right)\right)\left(\boldsymbol{h}_t\left(X_t,A_t,Y_t;R(\pi^{\mathrm{e}}),f_{t-1},\pi_\tau,\pi^{\mathrm{e}}\right)\right)^{\top}\xrightarrow{p}\Sigma$. The $(\tau,\tau)$-element of the matrix is

$$\frac{1}{T}\sum_{t=1}^{T}\frac{1}{r_\tau^2}\left(\sum_{a=1}^{K}\left\{\frac{\pi^{\mathrm{e}}(a\mid X_t)\mathbb{1}[A_t=a]\{Y_t-f_{t-1}(a,X_t)\}}{\pi_\tau(a\mid X,\Omega_{t_{\tau-1}})}+\pi^{\mathrm{e}}(a\mid X_t)f_{t-1}(a,X_t)\right\}-\theta\right)^2$$
$$\times\mathbb{1}\big[t_{\tau-1}<t\le t_\tau\big]$$

$$=\frac{1}{T}\sum_{t=1}^{T}\frac{1}{r_\tau^2}\left(\sum_{a=1}^{K}\left\{\frac{\pi^{\mathrm{e}}(a\mid X_t)\mathbb{1}[A_t=a]\{Y_t-f_{t-1}(a,X_t)\}}{\pi_\tau(a\mid X,\Omega_{t_{\tau-1}})}+\pi^{\mathrm{e}}(a\mid X_t)f_{t-1}(a,X_t)\right\}-\theta\right)^2$$
$$\times\mathbb{1}\big[t_{\tau-1}<t\le t_\tau\big]$$

$$-\frac{1}{T}\sum_{t=1}^{T}\frac{1}{r_\tau^2}\left(\sum_{a=1}^{K}\left\{\frac{\pi^{\mathrm{e}}(a\mid X_t)\mathbb{1}[A_t=a]\{Y_t-\mathbb{E}[Y(a)\mid X_t]\}}{\pi_\tau(a\mid X,\Omega_{t_{\tau-1}})}+\pi^{\mathrm{e}}(a\mid X_t)\mathbb{E}[Y(a)\mid X_t]\right\}-\theta\right)^2$$
$$\times\mathbb{1}\big[t_{\tau-1}<t\le t_\tau\big]$$

$$+\frac{1}{T}\sum_{t=1}^{T}\frac{1}{r_\tau^2}\left(\sum_{a=1}^{K}\left\{\frac{\pi^{\mathrm{e}}(a\mid X_t)\mathbb{1}[A_t=a]\{Y_t-\mathbb{E}[Y(a)\mid X_t]\}}{\pi_\tau(a\mid X,\Omega_{t_{\tau-1}})}+\pi^{\mathrm{e}}(a\mid X_t)\mathbb{E}[Y(a)\mid X_t]\right\}-\theta\right)^2$$
$$\times\mathbb{1}\big[t_{\tau-1}<t\le t_\tau\big].$$

The part

$$\frac{1}{T}\sum_{t=1}^{T}\frac{1}{r_\tau^2}\left(\sum_{a=1}^{K}\left\{\frac{\pi^{\mathrm{e}}(a\mid X_t)\mathbb{1}[A_t=a]\{Y_t-f_{t-1}(a,X_t)\}}{\pi_\tau(a\mid X,\Omega_{t_{\tau-1}})}+\pi^{\mathrm{e}}(a\mid X_t)f_{t-1}(a,X_t)\right\}-\theta\right)^2$$
$$\times\mathbb{1}\big[t_{\tau-1}<t\le t_\tau\big]$$

$$-\frac{1}{T}\sum_{t=1}^{T}\frac{1}{r_\tau^2}\left(\sum_{a=1}^{K}\left\{\frac{\pi^{\mathrm{e}}(a\mid X_t)\mathbb{1}[A_t=a]\{Y_t-\mathbb{E}[Y(a)\mid X_t]\}}{\pi_\tau(a\mid X,\Omega_{t_{\tau-1}})}+\pi^{\mathrm{e}}(a\mid X_t)\mathbb{E}[Y(a)\mid X_t]\right\}-\theta\right)^2$$
$$\times\mathbb{1}\big[t_{\tau-1}<t\le t_\tau\big]$$

converges in probability to $0$ because $f_{t-1}(a,X_t)\xrightarrow{\mathrm{P}}\mathbb{E}[Y(a)\mid X_t]$. The term

$$\frac{1}{T}\sum_{t=1}^{T}\frac{1}{r_\tau^2}\left(\sum_{a=1}^{K}\left\{\frac{\pi^{\mathrm{e}}(a\mid X_t)\mathbb{1}[A_t=a]\{Y_t-\mathbb{E}[Y(a)\mid X_t]\}}{\pi_\tau(a\mid X,\Omega_{t_{\tau-1}})}+\pi^{\mathrm{e}}(a\mid X_t)\mathbb{E}[Y(a)\mid X_t]\right\}-R(\pi^{\mathrm{e}})\right)^2$$
$$\times\mathbb{1}\big[t_{\tau-1}<t\le t_\tau\big].$$

converges in probability to

$$\frac{1}{r_\tau}\mathbb{E}\left[\sum_{a=1}^{K}\left\{\frac{\left(\pi^{\mathrm{e}}(a\mid X)\right)^2\mathrm{Var}\big(Y(a)\mid X\big)}{\pi_\tau(a\mid X,\Omega_{t_{\tau-1}})}+\left(\pi^{\mathrm{e}}(a\mid X)\mathbb{E}[Y(a)\mid X]-R(\pi^{\mathrm{e}})\right)^2\right\}\right].$$

from the weak law of large numbers for i.i.d. samples as $t_{\tau-1}-t_\tau\to\infty$ because the samples are i.i.d. between $t_{\tau-1}$ and $t_\tau$. $\qquad\square$

## C.3 OPE estimator when the true logging policy is unknown

Then, we consider estimating the policy value without using the true logging policy $\pi_t$. We use adaptive-fitting for obtaining an asymptotically normal estimator. As the ADR estimator under Assumption 2, we estimate $\pi_t$ and $f^*$ only using $\Omega_{t-1}$ and denote them as $g_{t-1}$ and $\hat{f}_{t-1}$, respectively.

Then, we define an estimator of $R(\pi^e)$ as

$$\tilde{R}_T^{\text{batch}}(\pi^e) = \arg\min_{R\in\mathbb{R}} \left(\tilde{\boldsymbol{q}}_T(R)\right)^\top \hat{W}_T \left(\tilde{\boldsymbol{q}}_T(R)\right),$$

where $\tilde{\boldsymbol{q}}_T(R) = \frac{1}{T}\sum_{t=1}^T \boldsymbol{h}_t\left(X_t, A_t, Y_t; R, \hat{f}_{t-1}, \hat{g}_{t-1}\pi^e\right)$ and $\hat{W}_T$ is a data-dependent $(M\times M)$-dimensional positive semi-definite matrix. This estimator is the ADR estimator under batch update. As well as the proof of Theorem 1, it is hold that

$$\left|\tilde{R}_T^{\text{batch}}(\pi^e) - \hat{R}_T^{\text{batch}}(\pi^e)\right| = \mathrm{o}_p(1/\sqrt{T})$$

if $\|\hat{g}_{t-1}(a|X_t) - \pi_{t-1}(a|X_t, \Omega_{t-1})\|_2 = \mathrm{o}_p(t^{-p})$, and $\|\hat{f}_{t-1}(a, X_t) - f^*(a, X_t)\|_2 = \mathrm{o}_p(t^{-q})$, where $p, q > 0$ such that $p + q = 1/2$. Therefore, we can obtain the following theorem.

**Theorem 5** (Asymptotic distribution the ADR estimator under batch update). *Suppose that*

**(i)** $\hat{W}_T \xrightarrow{\mathrm{p}} W$;

**(ii)** $W$ *is a positive definite.*

*Then, under Assumptions 1, 3, 4–6,*

$$\sqrt{T}\left(\tilde{R}_T^{\text{batch}}(\pi^e) - R(\pi^e)\right) \xrightarrow{\mathrm{d}} \mathcal{N}\left(0, \sigma^2\right),$$

*where $\sigma^2 = \left(I^\top WI\right)^{-1} I^\top W\Sigma W^\top I\left(I^\top WI\right)^{-1}$ and $\Sigma$ is a $(M\times M)$ diagonal matrix such that the $(\tau\times\tau)$-element is $\Psi(\pi^e, \pi_\tau)$.*

## D   Details of experiments

The description of the dataset is shown in Table 3. We use LinUCB and LinTS policies. We add uniform sampling to make overlap between policies. We can relax this requirement by considering different DGPs, such as batched sampling. However, for brevity, we adopt this setting. Additional results are shown as follows.

For numerical experiments in Section 6.2, we show the result with sample sizes $T = 100, 1,000$ in Table 4. In addition, we show the error distribution with sample size $T = 100$ in Figure 4; we show the error distribution with sample size $T = 500$ in Figure 5; we show the error distribution with sample size $T = 500$ in Figure 6; we show the error distribution with sample size $T = 1,000$ in Figure 7.

For experiments with dependent samples in Section 7, we show the additional results with different settings in Tables 5–10. In Table 5, we show the results using the benchmark datasets with 800 samples generated from the LinUCB algorithm. In Table 6, we show the results using the benchmark datasets with 1,000 samples generated from the LinUCB algorithm. In Table 7, we show the results using the the benchmark datasets with 1,200 samples generated from the LinUCB algorithm. In Table 8, we show the results using the the benchmark datasets with 800 samples generated from the LinTS algorithm. In Table 9, we show the results using the the benchmark datasets with 1,000 samples generated from the LinTS algorithm. In Table 10, we show the results using the the benchmark datasets with 1,200 samples generated from the LinTS algorithm.

Next, we compare the estimators using the benchmark datasets generated from the logistic regression as well as the evaluation weight; that is, the samples are i.i.d. For $\alpha \in \{0.7, 0.4, 0.1\}$ and the sample sizes 800, 1,000, and 1,200, we calculate the RMSEs and the SDs over 10 trials. We show additional results with different settings in Tables 11–13. In Table 11, we show the results using the benchmark datasets with 800 samples. In Table 12, we show the results using the benchmark datasets with 1,000 samples. In Table 13, we show the results using the benchmark datasets with 1,200 samples. In these experiments, AIPW estimators show better performances than the ADR estimator. We conjecture that this is because the logging policy is not unstable unlike the case with dependent samples; that is, the paradox is specific to the case where samples are dependent. Note this case (i.i.d. samples) is not in the scope of the proposed method. In such cases, it is common to use other methods, such as the cross-fitting proposed by Chernozhukov et al. (2018) instead of

the AIPW and ADR estimators discussed in this paper. We show this result to clarify the cause of the paradox. Since the ADR and AIPW estimators have the same asymptotic variance, and since the AIPW estimator uses more information, it is a natural result that the AIPW estimator performs better. However, when the samples are dependent, the ADR estimator paradoxically outperforms the AIPW estimator because of the unstable behavior of the logging policy $\pi_t$.

As a surprising discovery, our proposed ADR estimator shows better results than the AIPW estimator, although the AIPW estimator uses more information (true logging policy $\pi_t$) than the ADR estimator and their asymptotic properties are the same. As discussed above, we consider that this result is due to the logging policy's unstable behavior. Even when knowing the true logging policy $\pi_t$, we can stabilize estimation by reestimating the logging policy from $(A_t, X_t)$. This paradox is similar to the well-known property that the IPW estimator using an estimated propensity score shows a smaller asymptotic variance than the IPW using the true propensity score (Hirano et al., 2003; Henmi & Eguchi, 2004; Henmi et al., 2007). However, we consider that they are different phenomena. In previous studies, the paradox is mainly explained by differences in the asymptotic variance between the IPW estimators with the true and an estimated propensity score. On the other hand, for our case, the AIPW and ADR estimators have the same asymptotic variance, unlike IPW-type estimators; therefore, we cannot elucidate the paradox by traditional explanation.

Table 3: Specification of datasets

| Dataset | the number of samples | Dimension | the number of classes |
|---------|-----------------------|-----------|------------------------|
| mnist | 60,000 | 780 | 10 |
| satimage | 4,435 | 35 | 6 |
| sensorless | 58,509 | 48 | 11 |
| connect-4 | 67,557 | 126 | 3 |

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

| 0.7 | 0.060 | 0.002 | **0.005** | 0.0 | 0.012 | 0.000 | 0.009 | 0.000 | 0.378 | 0.013 |
| 0.4 | 0.043 | 0.002 | **0.010** | 0.0 | 0.014 | 0.000 | 0.023 | 0.000 | 0.211 | 0.012 |
| 0.1 | **0.017** | 0.000 | **0.017** | 0.0 | 0.037 | 0.002 | 0.035 | 0.001 | 0.057 | 0.003 |