# OpenReview forum: "The Adaptive Doubly Robust Estimator and a Paradox Concerning Logging Policy"
_NeurIPS.cc/2021/Conference — NeurIPS 2021 Poster_

### Official Review · Reviewer_goA7 · 2021-07-13

**Rating:** 7
**Confidence:** 3

**Summary:**

The authors propose and analyze an estimator for off policy value estimation (OPVE) in a Markovian setting where (X_t,A_t,Y_t(A_t)) ~ p(x)p_t(a|x)p(y_a|x) and the logging policy pi_t(a|x,Omega_{t-1}) is unknown. The estimator involves a familiar doubly robust moment function and a new form of cross fitting called adaptive fitting. The authors prove consistency, asymptotic normality, and semiparametric efficiency. Thorough simulations demonstrate low RMSE and correct coverage.

**Limitations And Societal Impact:**

What assumptions could be violated, and how would that impact conclusions and policy recommendations?

**Main Review:**

Overall assessment

Originality: The innovation is to allow the logging policy to be unknown. For the theory to extend to this case, adaptive fitting is introduced as an elegant solution that appears to be new. Please also compare with Kallus and Uehara’s mixingale sample splitting to shed light on originality.

Quality: The estimator and analysis are interesting. I am concerned about whether the Hadad et al estimator (2019) was given a good faith implementation since its coverage is zero (when it should be 95).

Clarity: This paper is unusually well written. See comments below about specific points to clarify or assumptions to further explain.

Significance: The contribution is clearly explained and not overstated. It is a natural progression of existing results that is relevant to empirical practice.

Suggestions to improve the paper

5: with non-Donsker

54: Nicely written

57: Nicely written

67: Nicely written

90: Crystal clear. The Markov property is a major simplification relative to dynamic treatment effects.

116: Please give further interpretation of Assumption 2.2. Unlike the other assumptions, it is not obvious. What cases are being ruled out?

145: the MSE to the true R(pi^e)? What does this mean given that R(pi^e) is a scalar for a given pi^e?

153: Impressively thorough

169: What is mixingale based sample splitting? How does it compare?

170: It would be nice to additionally mention the sample splitting of Chiang et al [arXiv:1909.03489]

180: please define MAB

264: What does this sentence mean?

Table 1: I am happy to see RMSE, SD, and CR

303: Please summarize the DGP in a clearer way, perhaps as an offset algorithm

328: It does not seem that the Hadad et al (2019) approach was given a good faith implementation. If their proposed weights do not include covariates, what would be a reasonable extension with covariates? Indeed, in Table 1, the coverage of AW-AIPW is even zero, which suggests this was not a reasonable implementation.

**Time Spent Reviewing:**

3

---

> ### Author Response · Authors · 2021-08-10
> **Response to Reviewer goA7**
>
> Thank you for your constructive comments and positive feedback.
> We will reflect your comments in the next update.
>
> ーーーーーーーーーーーーーーーーーーーーーーーーー
>
> ### Question 1
> **Q1(1)**. Please also compare with Kallus and Uehara’s mixingale sample splitting to shed light on originality.
>
> **Q1(2)**. 169: What is mixingale based sample splitting? How does it compare?
>
> ### Answer 1
> - Thank you for your important input. If a comparison is possible, it will be done in the next update. However, the situation considered in the Kallus and Uehara paper is different from our paper.
> - First, this is our typo, the method in Kallus and Uehara is based on the assumption of ``mixing,'' not ``mixingale.'' We will also fix our typo "Kallus & Uehara (2019) proposed a mixingale based sample-splitting" as "Kallus & Uehara (2019) proposed a sample-splitting with a mixing-based assumption."
> - Mixing does not imply martingale difference sequence, nor does martingale difference sequence imply mixing. For example, please refer to the following article:
> https://stats.stackexchange.com/questions/439076/does-a-martingale-difference-sequence-mds-imply-strong-mixing
> - We will carefully consider whether and under what circumstances the method in that paper can be adapted to our setting (martingale), and if so, we will conduct a comparison experiment.
> - In the sample-splitting of Kallus & Uehara, they first separate the whole dataset into several blocks without changing the order. For instance, for time-dependent samples $(W_1, W_2, \dots, W_T)$, they construct $n$ small groups $G_1 = (W_1,W_2,\dots, W_{t-1})$, $G_2 = (W_{t_1 + 1}, \dots, W_{t_2}), \dots G_n = (W_{t_{n-1}+1},\dots, W_T)$. The sample size of the blocks is assumed to increase as the overall sample size increases. Neighboring groups are strongly correlated, but under the assumption of mixing, the dependence between non-neighboring blocks is asymptotically negligible. By using this property, Kallus & Uehara (2019) proposed a method similar to cross-fitting of Chenozhukov et al. on non-adjacent blocks.
> - This technique is possible under the mixing assumption. However, under the martingale difference sequence assumption, the method cannot be applied. For example, such a sample splitting method does not allow the use of past samples when estimating the nuisance parameters using future data because we cannot condition the expectation of the past samples on the future information; that is, we cannot estimate the main parameter with $G_1$ by using nuisance parameters estimated from $G_3,\dots,G_{n}$. In this case, we expect that the convergence rate of the policy estimator is slower than the order of $sqrt{T}$ for a sample size $T$ because we need to skip the first block $G_1$. If the sample size of $G_1$ is $rT$, we can only attain $\sqrt{(1-r)T}$ convergence rate, but in our method we can easily obtain $\sqrt{T}$ convergence rate in our setting.
>
> ーーーーーーーーーーーーーーーーーーーーーーーーー
>
> ### Question 2
> **Q2(1)**. 116: Please give further interpretation of Assumption 2.2. Unlike the other assumptions, it is not obvious. What cases are being ruled out?
> **Q2(2)**. What assumptions could be violated, and how would that impact conclusions and policy recommendations?
>
> ### Answer 2
> - Assumption 2.2 requires that the logging policy cannot be $0$. Under this assumption, the optimal algorithm policy for regret minimization cannot be evaluated.
> - The two main applications of this paper are as follows:
> (i) Sub-optimal algorithm for regret minimization;
> (ii) Best-arm identification.
> - For instance, in industrial application, we often run several algorithms simultaneously, such as the uniform random sampling and Thompson sampling, to avoid the risk of unexpected deterioration in performance or to correct data. For instance, we apply the Thompson sampling to 95% of the samples and the uniform sampling to other 5% of the samples. In this case, the uniform sampling ensures the logging policy is larger than $0$, so we can apply our algorithm.
> - In best arm identification, it is optimal to assign a ratio of samples to each arm. This ratio is larger than $0$, so Assumption 2.2 holds for the optimal algorithm. For more details, see [3] Kaufmann et al. (2016) and [4] Garivier and Kaufmann (2017).
> [3] Kaufmann, Cappé, and Garivier. On the Complexity of Best Arm Identification in Multi-Armed Bandit Models. JMLR 2016.
> [4] Garivier and Kaufmann. Optimal Best Arm Identification with Fixed Confidence. COLT 2016.
> - To the best of our knowledge, there is no method that can be applied to the optimal algorithm in standard regret minimization setting even if we know the true logging policy.
>
> ーーーーーーーーーーーーーーーーーーーーーーーーー
>
> ### Question 3
> **Q3 (1)**. I am concerned about whether the Hadad et al estimator (2019) was given a good faith implementation since its coverage is zero (when it should be 95).
>
> **Q3 (2)**. 328: It does not seem that the Hadad et al (2019) approach was given a good faith implementation. If their proposed weights do not include covariates, what would be a reasonable extension with covariates? Indeed, in Table 1, the coverage of AW-AIPW is even zero, which suggests this was not a reasonable implementation.
>
> ### Answer 3
> - We consider that the implementation was not done in bad faith, and we made our best effort to give it a fair implementation. There are two possible reasons why the method of Hadad et al (2019) does not perform well.
> (i) The adaptive weight of Hadad et al (2019) causes a bias in the estimator;
> (ii) Hadad et al (2019) is designed for a case without covariates (context).
> Because the estimator is biased, if we misspecify the adaptive weight, the performance of the estimator can drop. In addition, even for cases without covariates, the choice of adaptive weight is not easy.
> - For these reason, it may not be appropriate to compare our method with the method of Hadad et al. (2019) in experiments even though the motivation is similar. In the next update, we will move the experiment results to Appendix.
>
> The following is a brief explanation of the complexities in this field.
> - The goal of the Hadad et al. (2019) paper seems to be twofold. One is empirical stabilization, and the other is to slightly expand the class of problem that the policy evaluation method can be applied. However, these does not seem to work well in the presence of covariates.
> - The work of Hadad et al. (2019) is a simplification of Ludtke and van der Laan (2016), which also considers stabilization in the presence of covariates and can handle more problems than Hadad. However, Ludtke and van der Laan (2016) requires sample splitting to estimate the stabilizing weights and cannot achieve a convergence rate of $\sqrt{T}$ for a sample size $\sqrt{T}$.
> - There are two major stabilization methods, including the work of Ludtke and van der Laan, that deal with covariates in a Hadad-like manner. All of them assume that a true logging policy can be used.
> - One is to estimate adaptive weights by sample splitting, such as Ludtke and van der Laan (2016). This approaches loses the convergence rate of $\sqrt{T}$, that is, they give the asymptotic normality with a slow rate.
> - The other methods are to put some constraints on the problem where we conduct policy evaluation to obtain a convergence rate of $\sqrt{T}$, such as the following work.
> [1] Zhan, Hadad, Hirshberg, and Athey. Off-Policy Evaluation via Adaptive Weighting with Data from Contextual Bandits
> [2] Bibaut, Chambaz, Dimakopoulou, Kallus, and van der Laan. Post-Contextual-Bandit Inference
> These were uploaded on arXiv at the same time this paper was submitted, so we will cite them in the next update. Our impression of these papers is that they can achieve a convergence rate of $\sqrt{T}$, but it is very difficult to check the assumptions for this method to be used. We conjecture that they can likely only be used in very limited situations.
> - The difficulty of this problem is that in order to achieve a convergence rate of $\sqrt{T}$, while estimating the adaptive weight, we need to limit the complexity of the logging policy in terms of the theories of empirical processes and semiparametric inference.
> - Thus, while such a stabilization can be a potentially powerful tool in situations where it is consistent with theory, the assumptions are difficult to check. Also, the motivation is similar but slightly different, making it difficult to make a one-to-one comparison with our method in experiments.
>
> ーーーーーーーーーーーーーーーーーーーーーーーーー
>
> ### Other comments
> **Comment　1**. 5: with non-Donsker
>
> **Reply 1**. Thank you for pointing out the typo. We will fix this.
>
> **Comment　2**. 90: The Markov property is a major simplification relative to dynamic treatment effects.
>
> **Reply 2**. We also consider that extension to a case with Markov property is an important work. However, we also think that this is slightly out of the scope of this paper.
>
> **Comment　3**. 145: the MSE to the true $R(pi^e)$? What does this mean given that $R(pi^e)$ is a scalar for a given $pi^e$?
>
> **Reply 3**. The MSE between an estimator $\hat{R}$ and $R(pi^e)$ is defined as:
> $$E[ (\hat{R} - R(pi^e))^2 ].$$
>
> **Comment　4**. 170: It would be nice to additionally mention the sample splitting of Chiang et al [arXiv:1909.03489]
>
> **Reply 4**. Thank you for your suggestion. We will cite the paper in the next update.
>
> **Comment　5**. 180: please define MAB
>
> **Reply 5**. We defined it in line 14.
>
> **Comments 6**. 264: What does this mean?
>
> **Reply 6**. This is our typo. We correct the sentence as ``We briefly introduce the ADR estimator with this case in Appendix C.''
>
> **Comments 7**.
> 303: Please summarize the DGP in a clearer way, perhaps as an offset algorithm
>
> **Reply 7**.
> We apologize, but do not quite understand what you are asking for.
> If the reviewer consider that making problem setting clearer will improve the presentation, we will add more explanations and figures in the next update.

---

> > ### Comment · Reviewer_goA7 · 2021-08-18
> > **Nice clarifications**
> >
> > Thanks for these thorough answers. I think that incorporating them into the paper will improve it. I have raised the score
> >
> > Allow me to clarify what I mean by "303: Please summarize the DGP in a clearer way, perhaps as an offset algorithm". As written, it is difficult to understand the data generating process used in the simulations. It may help to write how each variable is sampled in a step-by-step way offset from the main text: step 1: ..., step 2: ..., etc.

---

> > > ### Author Response · Authors · 2021-08-30
> > > **Re: Nice clarifications**
> > >
> > > We appreciate the reviewer raising the score and apologize for our delayed reply.
> > >
> > > > Allow me to clarify what I mean by “303: Please summarize the DGP in a clearer way, perhaps as an offset algorithm”. As written, it is difficult to understand the data generating process used in the simulations. It may help to write how each variable is sampled in a step-by-step way offset from the main text: step 1: ..., step 2: ..., etc.
> > >
> > > Thank you for clarifying your comment, and we agree with it.
> > > We will use the space gained in the additional page, which is allowed for the camera-ready version to update our introduction of the DGP as you suggested.
> > >
> > > Although we do not have much time left for discussion, please let us know if you have any further questions or comments.

---

### Official Review · Reviewer_Qkbm · 2021-07-17

**Rating:** 6
**Confidence:** 3

**Summary:**

In the setting studied in this paper, the analyst has access to data consisting of treatment assignments $A_t \\in \\mathcal{A} = \\{1,...,K\\}$, covariates $X_t$ and responses $Y_t = Y_t(A_t)$, where $(Y_t(a))_{a \\in \\mathcal{A}}$ are potential outcomes. The data are collected at time steps $t=1,...,T$ and the data generation occurs through an adaptive experiment. That is, at the $t$-th time step, the analyst can use all information up to time $t-1$, as well as $X_t$ and can decide with what probabilities $\\pi_t(a \\mid x)$ to assign any of the treatments in $\\mathcal{A}$ to the subject arriving at time $t$.   In other words, $A_t \\sim \\pi_t(\\cdot \\mid X_t)$ and then $Y_t = Y(A_t)$ is revealed.

The consequence of the adaptive experimentation is that the tuples $(X_t, A_t, Y_t)$ are dependent across time-steps $t$ (even though $(X_t, (Y_t(a))_{a \\in \\mathcal{A}})$ are iid and therefore it becomes a challenging task to conduct inference for average treatment effects (and other causal contrasts/estimands of interest).


Existing work for inference has focused on the situation wherein the policies $\\pi_t( \\cdot \\mid x)$, that were used during experimentation, are known to the analyst, while the response functions $f^*(a, x) = E[Y_t(a) \\mid X=x]$ are not known. The present paper develops a doubly robust approach to inference that also works when $\\pi_t( \\cdot \\mid x)$ is also not known to the analyst.

The paper also emphasizes that the new approach can sometimes outperform existing approaches that use additional information (on the true treatment assignment probabilities), without using the additional information (they call it a paradox).



**Limitations And Societal Impact:**

The authors discuss some limitations (but not all, see review above).

**Main Review:**


The paper is well-written and the assumptions are clearly stated. Adaptive experimentation is bound to become more popular, and most traditional statistical approaches break down in that setting. Instead new approaches are needed. Hence it is an important area of research, and this paper further contributes to this area.

1) The main aspect that is missing from the paper, in my opinion, is some motivation for the statistical setting. I find it hard to imagine that someone has access to data from an adaptive experiment (and e.g., the temporal succession of treatment assignments $t=1,...,T$ is known), but does not know how the experiment was actually implemented (i.e., how the treatment assignments were sequentially randomized). When would this occur? This is in contrast to e.g., the typical setting for doubly robust estimators for estimating average treatment effects in observational data. There, the treatment assignment is clearly not known and one estimates propensities and potential outcome regression functions to deal with confounding (under weaker conditions compared to both the IPW approach and the outcome modeling approach).

2) I wonder whether the novelty of the "adaptive fitting" procedure introduced by the authors may be overstated. For example, Hadad et al. also fit the outcome models $f^*(a, x)$ based on data up to the previous time point (similar to the adaptive fitting in this paper), so that I am not convinced of the novelty of this methodological contribution. I do agree however with the authors, that it appears that from a technical perspective, also having an estimated $\\hat{\\pi}_t(\\cdot \\mid X_t)$ based on past data (and not just response function) would make the theoretical analysis more challenging.

3) What is the take home message from the "paradox"? Can the authors provide practical recommendations about what to do in case the treatment assignment probabilities are actually known?


Some minor remarks: On Page 6 there is typo "wit" instead of "with". I also wanted to let the authors know that  Zhan et al. (2021) have a follow-up on the paper of Hadad et al. that accounts for context covariates (https://arxiv.org/pdf/2106.02029.pdf). (Of course the reviewed manuscript precedes the preprint I linked above, so I just provided the above reference in case it is useful, the authors do not need to cite it or compare to it.)

**Update:** Based on the author response, one take-away is that the proof techniques and technical results are a key contribution of this paper. I have therefore increased my score from 5 to 6. I am still not entirely convinced however about the practicality of the statistical setting (i.e., I am not convinced that a tech company doing adaptive experimentation would have trouble storing the experimental protocol).


**Time Spent Reviewing:**

4

---

> ### Author Response · Authors · 2021-08-09
> **Response to Reviewer Qkbm**
>
> Thank you for your insightful comments.
>
> The reviewer raises an important question on the practicality of assuming a situation in which the true logging policy is not known in an adaptive experimental design. While the reviewer correctly points out that it is the case that the true logging policy is often known in real applications, the true logging policy tends to be difficult to store or compute in the adaptive experiment setting. Thus, even when the true logging policy can be known, it can be difficult to be used. For example, take for instance the usage of Thompson sampling, which selects the most rewarding arm amongst the arms sampled from the posterior distribution. Since the algorithm only chooses the sampled arm, the probability that an arm was chosen is not recorded in the experiment. In order to calculate the logging policy, we need to compute the posterior distribution, either analytically or via Monte Carlo simulation. Either way, we need to store the same number of models as the sample in the adaptive experiment, since the policy is updated in each sample, and also the calculation can be very costly.  Thus, the usefulness of the proposed method does not only hold for when the logging policy is unknown, but holds for when the logging policy is infeasible to attain or store, reducing cost. Additionally, the paradox we report suggests that, even if the true logging policy is known, the proposed method may improve performance.
>
> In the next update, we will reflect your comments.
> Detailed replies are as follows.
>
> ーーーーーーーーーーーーーーーーーーーーーーーーー
>
> ### Question 1
> The main aspect that is missing from the paper, in my opinion, is some motivation for the statistical setting. I find it hard to imagine that someone has access to data from an adaptive experiment (and e.g., the temporal succession of treatment assignments  is known), but does not know how the experiment was actually implemented (i.e., how the treatment assignments were sequentially randomized). When would this occur? This is in contrast to e.g., the typical setting for doubly robust estimators for estimating average treatment effects in observational data. There, the treatment assignment is clearly not known and one estimates propensities and potential outcome regression functions to deal with confounding.
>
> ### Answer 1
> - As the reviewer pointed out, in many experimental designs, the experiment can be manipulated by the experimenter, but the logging policy is often not recorded because it is costly in adaptive experiment, since we need to preserve a policy for each sample.
> - For example, let us consider Thompson sampling, where an algorithm selects the best arm sampled from the posterior distribution. In this case, if we want to know the probability of choosing a treatment (logging policy), we need to calculate the probability that each arm is chosen from the posterior distribution. The calculation of this probability is not necessary for the bandit algorithm itself, but is an additional task for policy evaluation.
> - In addition, to compute the probability (logging policy), we not only need to record the information of the posterior distribution, but also need to calculate the probability by Monte Carlo simulation from the posterior distribution, if no analytical solution is available. This is a difficult task in terms of both computational volume and computational resources.
> - Moreover, in adaptive experiments in industry, such as AB tests, the number of logs is often in the order of several million. In such cases, storing the logging policy is also a difficult task.
> - While we may be able to know the logging policy, if we make a significant effort, for policy evaluation, such an effort may be unnecessary. This is because, as we have shown in this manuscript, not only can we know asymptotic normality without knowing the logging policy, but also, as the paradox indicates, the performance may be better if we do not use the logging policy directly.
> - We have avoided mentioning this area in the paper because different practitioners may have different opinions on the matter. However, as the reviewer pointed out, it is of interest to readers, and we will introduce such considerations in the next update.
>
>
> ーーーーーーーーーーーーーーーーーーーーーーーーー
>
> ### Question 2
> I wonder whether the novelty of the "adaptive fitting" procedure introduced by the authors may be overstated. For example, Hadad et al. also fit the outcome models  based on data up to the previous time point (similar to the adaptive fitting in this paper), so that I am not convinced of the novelty of this methodological contribution. I do agree however with the authors, that it appears that from a technical perspective, also having an estimated based on past data (and not just response function) would make the theoretical analysis more challenging.
>
> ### Answer 2
> - We believe that we have made two major methodological contributions compared to the previous literature:
> (i) not relying on the martingale property to show asymptotic normality;
> (ii) not using Chernozhukov's cross-fitting, for semiparametric inference, even when the samples are dependent.
> Though these two conditions are almost always relied upon in previous research, we consider both conditions to be problematic in the setting we consider in our paper. As far as we are aware, this paper is the first to relax both conditions, which we hope expands the applicability of this approach.
> - For example, for (ii), the following paper also tries to use doubly robust estimator in adaptive setting with a variant of cross-fitting.
> Waudby-Smith, Arbour, Sinha, Kennedy, and Ramdas. Doubly robust confidence sequences
> for sequential causal inference (arXiv)
> They propose ``sequential sample splitting'' in which they separate the whole samples to two groups during an adaptive experiment: evaluation and training group. In the training group, they estimate the nuisance parameters, and in the evaluation group, they estimate the semiparametric policy value using estimated nuisance parameters. This is not an off-policy setting, but an on-policy setting, which allows such an operation. In addition, it seems that they do not use training samples for evaluation owing to the sample dependency; that is, discarding a part of samples in statistical inference, unlike standard cross-fitting or our adaptive-fitting. Based on our findings, this may not be necessary, since we can just sample split step-wise, even when the samples are dependent. Although their motivation is different from ours, they may not be aware that sample splitting, done our way, is possible. Thus, we consider that our proposed method will improve those methods potentially and is worth reporting as non-trivial.
> (The goal of that paper seems to be different from ours. We will cite it in the next update.)
> - Most studies in this field rely on the martingale property, and are limited to that problem setting. Our findings remove this limitation and expands the applicability significantly. For example, insistence on using a true logging policy would degrade experimental performance in cases where the logging policy is close to zero.
> - In addition, the framework of this method and the method in Hadad et al. (2019) cannot be applied to problems where actions are chosen with probability $1$ ($\pi_t(a| x, \Omega_{t-1}) = 1$ for an action $a$, and $\pi_t(a'| x, \Omega_{t-1}) = 0$ for the other actions $a'$). This is because in such a case, the logging policy of a particular action will be $1$ and the rest will be $0$, and the IPW part of the doubly robust estimator will diverge infinitely by dividing by $0$. However, this is a common setting for bandits. Even in such a case, policy evaluation may be possible by replacing the true logging policy containing $\pi_t(a'| x, \Omega_{t-1}) = 0$ with other values, such as the asymptotic allocation ratio $\lim_t \sum^t_{s=1}\pi_t(a'| x, \Omega_{t-1}) = \alpha(a' | x) > 0$. This is a future research of this work, but we are already working on this problem for this particular bandit case. Since we cannot present the work for double blind review, we send the confidential message to the Area Chair to explain the details.
>
> ーーーーーーーーーーーーーーーーーーーーーーーーー
>
> ### Question 3
> What is the take home message from the "paradox"? Can the authors provide practical recommendations about what to do in case the treatment assignment probabilities are actually known?
>
> ### Answer 3
> Our main take home messages are as follows:
> - Policy evaluation can be done without knowing the true logging policy, so if storing or calculating the logging policy is costly, there is no need to do so. There is even a possibility that the performance can be improved by not using it.
> - When we know the true logging policy, we often use the true logging policy + weight-clipping. However, in the case of adaptive experiments, if the behavior of the logging policy is unstable in the early rounds and then stabilizes in the later rounds, we can use an estimator that works better experimentally, without sticking to the true logging policy.
>
> In summary, the take home message is that this paradox allows us to avoid calculating and storing a logging policy and also allows us to utilize the statistics in a flexible way that we can look for a better experimental estimator even when we know the true logging policy.
>
> ーーーーーーーーーーーーーーーーーーーーーーーーー
>
> ### Other comments
> I also wanted to let the authors know that Zhan et al. (2021) have a follow-up on the paper of Hadad et al. that accounts for context covariates. (Of course the reviewed manuscript precedes the preprint I linked above, so I just provided the above reference in case it is useful, the authors do not need to cite it or compare to it.)
>
> Thank you for sharing this interesting paper. It was uploaded to arXiv at the same time as the submission, so we could not cite it. We have already read the paper and are checking their proofs. We plan to cite it in the next update.

---

### Official Review · Reviewer_sEtk · 2021-07-18

**Rating:** 7
**Confidence:** 3

**Summary:**

The paper studies off-policy evaluation from data generated from unknown and adaptive logging policies (logging policy can be updated based on past observations). The paper modified the doubly robust estimator in the non-adaptive and known-logging-policy setting to this setting by using an estimated logging policy and an estimated reward model. The estimation models used at each time step are built only on data collected before. They show the asymptotic normality of this estimator under some assumptions and make the step-wise construction and how it achieves asymptotic normality as a general approach. They empirically evaluated the proposed estimator on some simulation dataset and  discussed why the proposed estimator enjoys smaller mean squared error empirically compared to estimators with known logging policy information, even when they all have asymptotic normality.

**Limitations And Societal Impact:**

Yes

**Main Review:**

Off-policy evaluation with unknown and adaptive logging policies is pretty realistic setting in real-world applications.

The paper propose an off-policy evaluation estimator in this setting and characterize the conditions under which it can achieve asymptotic normality, which provides theoretical groundings for using this estimator.

One question I have is, if we apply adaptive-fitting to DM and EIPW estimators, do we achieve asymptotic normality. And it would be great to also include the experiments comparing with these two baselines as reference.


Minor issues:

The acronym AIPW is very confusing. I do not know what 'A' stands for and I always thought it was adaptive when I saw it. But it was not.

line 226 satisfies what?

line 360 as section 6.3 -> as discussed in Section 6.3


**Time Spent Reviewing:**

4

---

> ### Author Response · Authors · 2021-08-09
> **Response to Reviewer sEtk**
>
> Thank you for your constructive comments. We will reflect your comments on our manuscript in the next update.
>
> Our replies to your comments are shown below.
>
> ーーーーーーーーーーーーーーーーーーーーーーーーー
>
> ### Question
> One question I have is, if we apply adaptive-fitting to DM and EIPW estimators, do we achieve asymptotic normality. And it would be great to also include the experiments comparing with these two baselines as reference.
>
> ### Answer
> - We cannot show the asymptotic normality for these two estimators.
> - This is because the convergence of the bias that vanished asymptotically is insufficient when using the adaptive-fitting to the DM and EIPW estimators.
> - In a few parts of the paper, we make slight mentions, e.g. Section 5 and Appendix C. However, we did not explain that it could not be estimated explicitly. In the next update, we will note that we cannot show the asymptotic normalities of the DM and EIPW estimators even when using adaptive-fitting.
> - We explain the reason why we cannot show the asymptotic normality in more details as follows. In the DR estimator, the estimator is given as,
> $$ \frac{1}{T}\sum^T_{t=1}\sum^K_{a=1}\pi^e(a | X_t)\left(\frac{1\[A_t = a\](Y_t(a) - \hat{E}[Y_t(a) | X_t ])}{\hat\pi_t(a | X_t, \Omega_{t-1})} + \hat{E}[Y_t(a) | X_t ]\right), $$
> where $ \hat{}$ denotes that it is an estimator. On the other hand, the DM estimator considers
> $$\frac{1}{T}\sum^T_{t=1}\sum^K_{a=1}\pi^e(a | X_t)\hat{E}[Y_t(a) | X_t ],$$
> and the EIPW estimator considers
> $$ \frac{1}{T}\sum^T_{t=1}\sum^K_{a=1}\pi^e(a | X_t)\frac{1\[A_t = a\]Y_t(a)}{\hat\pi_t(a | X_t, \Omega_{t-1})}.$$
> The convergence rates of the standard estimators cannot exceed $O_p(1/\sqrt{T})$. For instance, the convergence rate of a sample mean of some random variables is $O_p(1/\sqrt{T})$. More complicated estimators have slower convergence rates. Under the convergence rate larger than $O_p(1/\sqrt{T})$, the bias affects the asymptotic distribution. For example, the DM estimator has the bias such that
> $$\left|\frac{1}{T}\sum^T_{t=1}\sum^K_{a=1}\pi^e(a | X_t)\hat{E}[Y_t(a) | X_t ] - \frac{1}{T}\sum^T_{t=1}\sum^K_{a=1}\pi^e(a | X_t)E[Y_t(a) | X_t ] \right|,$$
> and the EIPW estimator has the bias such that
> $$\left| \frac{1}{T}\sum^T_{t=1}\sum^K_{a=1}\pi^e(a | X_t)\frac{1\[A_t = a\]Y_t(a)}{\hat\pi_t(a| X_t, \Omega_{t-1})} - \frac{1}{T}\sum^T_{t=1}\sum^K_{a=1}\pi^e(a | X_t)\frac{1[A_t = a]Y_t(a)}{\pi_t(a| X_t, \Omega_{t-1})}\right|.$$
> Here, $\lVert E[Y_t(a) | X_t ] - \hat{E} [Y_t(a) | X_t ] \rVert_2$ and $\lVert \pi_t(a| X_t, \Omega_{t-1}) - \hat\pi_t(a| X_t, \Omega_{t-1})\rVert_2$ larger than or equal to $O_p(1/\sqrt{T})$, so the bias does not vanish with $o_p(1/\sqrt{T})$, which is required to show the asymptotic normality. In contrast, in the DR estimator, the bias depends on
> $$\lVert E[Y_t(a) | X_t ] - \hat{E}[Y_t(a) | X_t ]\rVert_2 \lVert \pi_t(a  | X_t, \Omega) -  \hat\pi_t(a  | X_t, \Omega) \rVert_2. $$
> Therefore, for example, if $\lVert E[Y_t(a) | X_t ] - \hat{E}[Y_t(a) | X_t ]\rVert_2 = o_p(1/T^{1/4})$ and $\lVert pi_t(a| X_t, \Omega_{t-1}) - \hat\pi_t(a| X_t, \Omega_{t-1})\rVert_2  = o_p(1/T^{1/4})$, the bias decays with $o_p(1/\sqrt{T})$. Therefore, only for the DR estimator, we can show the asymptotic normality.
> - The reviewer also can refer Chernozhukov et al 2016 for this problem, in which they discuss the semiparmetric inference with i.i.d. samples
> - Even for i.i.d. samples, we can show the asymptotic normalities only for very limited cases.
> - For instance, [1] Hirano, Imbens, and Ridder (2003) showed the asymptotic normality of the EIPW estimator when we estimate the true logging policy with the "sieve'' estimator, such as regression with the power series.
> [1] Hirano, Imbens, and Ridder. Efficient Estimation of Average Treatment Effects Using the Estimated Propensity Score. Econometrica 2003.
>
>
> ーーーーーーーーーーーーーーーーーーーーーーーーー
>
> ### Other comments
> **Comment 1**. The acronym AIPW is very confusing. I do not know what 'A' stands for and I always thought it was adaptive when I saw it. But it was not.
>
> **Reply 1**. The name of the AIPW estimator followed the custom in the case with i.i.d. samples. In the next update, we may change the name to Adaptive AIPW (A2IPW) estimator or add some emphasis on the name.
>
> **Comment 2**. line 226 satisfies what?
>
> **Reply 2**. This is our typo. We will fix "satisfies $E[\psi(W_t; \theta_0, \eta_0)]$," to
> "satisfies $E[\psi(W_t; \theta_0, \eta_0)] = 0$."

---

> > ### Comment · Reviewer_sEtk · 2021-08-28
> > **Thanks for the explanation**
> >
> > Thank the authors for the clear explanation on whether we can achieve asymptotic normality by applying adaptive-fitting to DM and EIPW estimators. It would be great to discuss this in the paper. I would like to raise my evaluation to 7.

---

> > > ### Author Response · Authors · 2021-08-30
> > > **Re: Thanks for the explanation**
> > >
> > > We would like to thank the reviewer for increasing the score.  We apologize for our delay in reply.
> > > The topic that the reviewer asked us was discussed around 2000 for i.i.d. samples. We will explain the background in more detail by citing the relevant literature.
> > > If the reviewer has any other questions, please ask us.

---

### Decision · Program_Chairs · 2021-09-27

**Decision:**

Accept (Poster)

**Comment:**

The paper studies off policy estimation using data collected by a changing (adaptive) and unknown logging policy. This is an important and under-explored setting, since adaptive experimentation is increasingly common in many application domains (e.g., online platforms). The paper gives an adaptive doubly-robust estimator, establishes semiparametric efficiency of the estimator, and also provides some nice simulation studies that showcase how fitting propensities may be beneficial even when the true ones are known.

Overall, all of the reviewers were quite positive about the paper, and so we are recommending acceptance. Please incorporate comments from the discussion section into the final version, as this will clear up some confusions and improve the manuscript.